# CADO: Cost-Aware Diffusion Models for Combinatorial Optimization via RL Finetuning

## Abstract

Recent advancements in Machine Learning (ML) have demonstrated significant potential in addressing Combinatorial Optimization (CO) problems through data-driven approaches. Heatmap-based methods, which generate solution heatmaps in a single step and employ an additional decoder to derive solutions for CO tasks, have shown promise due to their scalability for large-scale problems. Traditionally, these complex models are trained using imitation learning with optimal solutions, often leveraging diffusion models. However, our research has identified several limitations inherent in these imitation learning approaches within the context of CO tasks. To overcome these challenges, we propose a 2-phase training framework for diffusion models in CO, incorporating Reinforcement Learning (RL) fine-tuning. Our methodology integrates cost information and the post-process decoder into the training process, thereby enhancing the solver's capacity to generate effective solutions. We conducted extensive experiments on well-studied combinatorial optimization problems, specifically the Traveling Salesman Problem (TSP) and Maximal Independent Set (MIS), ranging from small-scale instances to large-scale scenarios. The results demonstrate the significant efficacy of our RL fine-tuning framework, surpassing previous state-of-the-art methods in performance.

## 1  Introduction

Combinatorial optimization (CO) has long been studied in operations research and computer science, but its inherent complexity, particularly the NP hardness, makes finding optimal solutions challenging (Karp, 1975). Typically, problem-specific heuristics (Papadimitriou & Steiglitz, 1998), such as the 2OPT heuristic for the traveling salesman problem, achieve high performance but lack generalization. Exact solvers also exist, but suffer from exponential growth in complexity with larger tasks. Recently, Machine Learning (ML) has shown significant potential to overcome these limitations, improving scalability for real-world applications (Bello et al., 2016; Vinyals et al., 2015; Khalil et al., 2017; Bengio et al., 2021). Constructive ML-based CO solvers, which generate solutions directly through neural networks, are emerging as promising alternatives to traditional heuristics (Zheng et al., 2024; Sun & Yang, 2023).

Research on these constructive solvers for CO is categorized as follows: autoregressive and heatmap-based models (non-autoregressive models). Autoregressive models generate solutions sequentially applying neural network feedforward operations, extending partial solutions step by step until an entire solution is formed (da Costa et al., 2020; Wu et al., 2019; Kool et al., 2019b; Kwon et al., 2020; Kim et al., 2022). However, these repeated feedforward operations lead to increased computational overhead and potential instability during both training and inference, limiting their efficiency in large-scale problems. In contrast, heatmap-based models generate an entire solution in a single feedforward pass by producing a heatmap that represents the probability that the edges or nodes are part of the solution (Fu et al., 2021a; Geisler et al., 2022; Joshi et al., 2019a). The generated heatmaps are then decoded into valid discrete CO solutions by using simple decoders such as greedy decoding algorithm.

Heatmap-based models frequently necessitate high-dimensional outputs, which in turn require complex models with numerous parameters. To ensure stable training of these intricate models, supervised learning (SL) is generally preferred. In this approach, the solver aims to generate heatmaps that

imitate high-quality solutions (Fu et al., 2021a; Geisler et al., 2022; Joshi et al., 2019b). Recently, powerful generative models—successful in image and language domains—have been applied as CO solvers (Graikos et al., 2022a; Mirhoseini et al., 2021; Kool et al., 2019a; Niu et al., 2020; Sun & Yang, 2023). However, we have identified several issues with existing SL-based heatmap solvers that merely focus on imitating optimal solutions. First, simply imitating optimal solutions does not always guarantee high-quality solution in CO. Second, the impact of decoding strategies on final solution quality is ignored during training. Third, these solvers heavily depend on high-quality training datasets, which are computationally intractable to obtain for large-scale CO problems due to their NP-hardness.

To address these issues, we introduce a reinforcement learning (RL) fine-tuning framework that directly incorporates cost information during training, specifically focusing on diffusion models. Our proposed approach, CADO (**C**ost-**A**ware **D**iffusion solver for combinatorial **O**ptimization), preserves the advantages of existing supervised learning (SL)-based solvers while effectively minimizing the cost of decoded solutions, the fundamental objective of combinatorial optimization problems. We selected diffusion models as our base architecture due to their recent successes across various domains, including combinatorial optimization problems (Sun & Yang, 2023; Li et al., 2023), as well as their inherent compatibility with RL. Through our effective RL fine-tuning process, our method successfully addresses the three aforementioned limitations of SL-based heatmap solvers, demonstrating substantial and consistent performance improvements across diverse combinatorial optimization tasks.

Our contributions can be summarized as follows: 1) We identify three issues of existing SL-based heatmap solvers that arise mainly from ignoring cost information during CO. 2) We introduce an RL fine-tuning algorithm for diffusion models that incorporates cost information in CO, along with practical techniques for stable RL fine-tuning. 3) Despite its simplicity, CADO demonstrates its strong effectiveness to address all three identified issues. 4) Building on these advantages, CADO outperforms existing algorithms across diverse CO benchmarks - including the Traveling Salesman Problem (TSP) with node sizes ranging from 50 to 10,000, the Maximum Independent Set (MIS), and the TSPLIB real dataset benchmark.

## 2 PRELIMINARIES

### 2.1 PROBLEM FORMULATION

We define the problem and introduce the key notations related to combinatorial optimization (CO) problems. Let $\mathcal{G}$ be the set of all CO instances, and let $g \in \mathcal{G}$ denote an instance. Each instance $g$ has an associated discrete solution space $\mathcal{X}_g := \{0,1\}^{N_g}$ and an objective function $c_g : \mathcal{X}_g \to \mathbb{R}$ for each solution $x \in \mathcal{X}_g$ defined as:

$$c_g(\boldsymbol{x}) = \text{cost}(\boldsymbol{x}, g) + \text{valid}(\boldsymbol{x}, g). \tag{1}$$

Here, $\text{cost}(\cdot)$ represents the cost value to be optimized, while $\text{valid}(\cdot)$ is a constraint indicator function, where $\text{valid}(\boldsymbol{x}, g) = 0$ if the solution $\boldsymbol{x}$ belongs to the feasible solution space $\mathcal{F}_g \subset \mathcal{X}_g$, and $\text{valid}(\boldsymbol{x}, g) = \infty$ when $\boldsymbol{x} \notin \mathcal{F}_g$. The optimization goal is to find the optimal solution $\boldsymbol{x}_\star$ for a given instance $s$:

$$\boldsymbol{x}_\star^g = \underset{\boldsymbol{x} \in \mathcal{X}_g}{\arg\min} \, c_g(\boldsymbol{x}). \tag{2}$$

We describe two specific CO problems as examples: the Traveling Salesman Problem (TSP) and the Maximal Independent Set (MIS) problem. In the TSP, an instance $g$ represents the coordinates of $n$ cities to be visited. The solution $\boldsymbol{x}$ is an $n \times n$ matrix, where $\boldsymbol{x}[i, j] = 1$ if the traveler moves from city $i$ to city $j$. The total solution space is $\mathcal{X}_g = \{0,1\}^{n \times n}$, and the feasible solution space $\mathcal{F}_g \subset \mathcal{X}_g$ is the set of all feasible TSP tours that visit each city exactly once. The objective function $\text{cost}(\cdot)$ represents the total length of the given tour and should be minimized. In the MIS problem, an instance $g$ represents a graph $(V, E)$, where $V$ is the vertex set and $E$ is the edge set. The solution space $\mathcal{X}_g = \{0,1\}^V$ indicates whether each vertex $v \in V$ is included in the solution set. To satisfy the independence property, $\boldsymbol{x}$ should not contain nodes connected by edges in $E$. The objective function $\text{cost}(\cdot)$ represents the total number of selected nodes and should be maximized.

## 2.2 Neural combinatorial optimization solver

In this section, we briefly introduce the concepts of autoregressive solvers and heatmap-based solvers. Autoregressive solvers extend a partial solution until a complete solution is formed:

$$p_\theta(\boldsymbol{x}|g) = \prod_{t=1}^{N_g} p_\theta(x_t|\boldsymbol{x}_{1:t-1}, g) \tag{3}$$

where $\boldsymbol{x}_{1:t-1}$ is the partial solution. This approach works very well for small-scale combinatorial optimization (CO) problems, but it becomes less practical for larger scales due to quadratic time and space complexity.

The heatmap-based solvers, proposed to effectively solve large-scale CO problems, directly generate a heatmap $\mathcal{H} \in \mathbb{R}^{N_g}$ , representing the likelihood of each variable being part of the solution, and utilizes it to form the final solution through an additional post-process decoder $p(\boldsymbol{x}|\mathcal{H})$ :

$$p_\theta(\boldsymbol{x}|g) = p_\theta(\mathcal{H}|g)p(\boldsymbol{x}|\mathcal{H}) \tag{4}$$

Simply sampling a solution from $\mathcal{H}$ may result in solutions belonging to the total solution space $\mathcal{X}_g$, which means that feasibility is not guaranteed. Therefore, heatmap-based solvers necessarily require a post-process decoder to convert solutions in the infeasible space to the feasible space $\mathcal{X}_g$. Various post-process decoders have been proposed. For example, Qiu et al. (2022) employs a method that stochastically samples valid variables while masking away infeasible variables, while Sun & Yang (2023) deterministically adds variables to the partial solution in descending order of the heatmap value as long as no conflicts occur.

Each solver can be trained using either a reinforcement learning (RL) or a supervised learning (SL) objective. In SL, the availability of high-quality solutions $\boldsymbol{x}_g^\star$ for each training instance $g$ are assumed to be given. The objective of SL is to maximize the likelihood :

$$\mathcal{L}(\theta) = \mathbb{E}_{g \sim \boldsymbol{P}(g)}[-\log p_\theta(\boldsymbol{x}_\star^g|g)]. \tag{5}$$

In RL, the solver does not assume the availability of the high-quality solutions $\boldsymbol{x}_\star^g$ for a given instance $g$. However, the solver exploits the information of the objective function $c_g(\cdot)$ during exploration and exploitation of the solutions $\boldsymbol{x}$. The objective of RL is to find a distribution $p_\theta(\boldsymbol{x}|g)$ that maximize the reward (minimize the cost) :

$$\mathcal{R}(\theta) = \mathbb{E}_{g \sim \boldsymbol{P}(g), \, \boldsymbol{x} \sim p_\theta(\boldsymbol{x}|g)}[-c_g(\boldsymbol{x})]. \tag{6}$$

## 2.3 Diffusion model for CO

Sun & Yang (2023) propose a diffusion model-based CO solver called DIFUSCO, which is classified as a heatmap-based solver. In DIFUSCO, the solver $p_\theta(\mathcal{H}|g)$ is modeled as a diffusion model and trained in a supervised manner.

The diffusion process consists of a forward noising procedure and a reverse denoising procedure. During the forward process, noise is gradually added to the initial solution until the solution is completely transformed into random noise, creating a sequence of latent variables $\mathbf{x_0}, \mathbf{x_1}, \ldots, \mathbf{x_T}$ where $\mathbf{x_0} = \boldsymbol{x}_\star^g$ in CO and $\mathbf{x}_T$ is completely random noise. The forward noising process is defined by $q(\mathbf{x_{1:T}}|\mathbf{x_0}) = \prod_{t=1}^{T} q(\mathbf{x_t}|\mathbf{x_{t-1}})$. Then, during the reverse denoising procedure, a model is trained to restore this random noise $\mathbf{x_T}$ back to the high-quality solution $\mathbf{x_0}$. The reverse process is modeled as $p_\theta(\mathbf{x_{0:T}}|g) = p(\mathbf{x_T}) \prod_{t=1}^{T} p_\theta(\mathbf{x_{t-1}}|\mathbf{x_t}, g)$, with $\theta$ representing the model parameters, and this reverse model is later used as a heatmap-based solver.

The training objective is to match $p_\theta(\mathbf{x_0}|g)$ with the high quality data distribution $q(\mathbf{x_0}|g)$, optimized by minimizing the variational upper bound of the negative log-likelihood:

$$\mathcal{L}(\theta) = \mathbb{E}_q\Big[-\log p_\theta(\mathbf{x_0}|\mathbf{x_1}, g) + \sum_{t=2}^{T} D_{KL}(q(\mathbf{x_{t-1}}|\mathbf{x_t}, \mathbf{x_0})\|p_\theta(\mathbf{x_{t-1}}|\mathbf{x_t}, g))\Big]. \tag{7}$$

More details are described in Appendix A.

## 3 MOTIVATION: ISSUES IN SL-BASED HEATMAP SOLVERS

Several supervised learning (SL)-based heatmap solvers have emerged in combinatorial optimization (CO) (Nowak et al., 2018; Joshi et al., 2019a; Fu et al., 2021a; Geisler et al., 2022; Sun & Yang, 2023). These approaches are trained on high-quality solution datasets, treating existing solutions as ground-truth labels to generate corresponding heatmaps. Their underlying assumption is that heatmaps closely approximating optimal solutions will be naturally decoded into high-quality, low-cost solutions. However, our analysis reveals several non-trivial issues that impact their effectiveness as CO solvers. In this section, we identify three fundamental challenges in training SL-based heatmap solvers for CO problems.

### 3.1 IGNORANCE OF COST IN TRAINING PROCESS

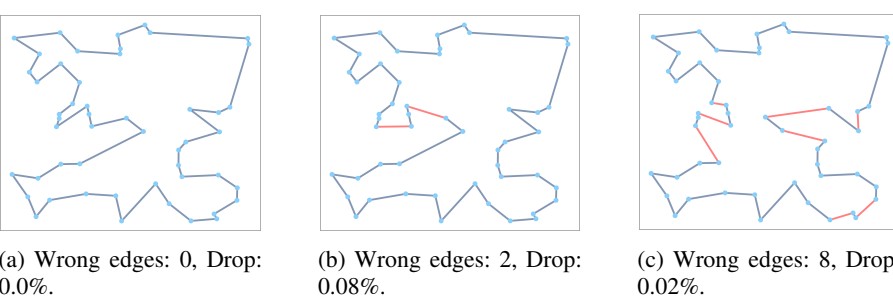

(a) Wrong edges: 0, Drop: 0.0%.

(b) Wrong edges: 2, Drop: 0.08%.

(c) Wrong edges: 8, Drop: 0.02%.

Figure 1: Examples of solutions with higher prediction error but lower cost compared on TSP-50.

We observe that a solution that is very similar to the optimal solution does not always guarantee a very low cost. To illustrate this point, consider the example shown in Figure 1. Suppose that we are given an optimal tour (a) that we want to predict. During training, the model generates two solutions: tour (b) and tour (c). Since tour (b) has fewer incorrectly connected edges, the SL training objective is likely to prefer tour (b) over tour (c). However, in terms of actual cost, tour (c) performs better than tour (b), revealing the inadequacy of the current training objective in properly distinguishing them. Therefore, cost information must be considered during training to achieve our true goal in CO, which is to minimize the cost.

### 3.2 IGNORANCE OF DECODERS IN TRAINING PROCESS

As we mentioned in Section 2.1, the feasible solution space $\mathcal{F}_g$ is a much smaller subset compared to the total solution space $\mathcal{X}_g$. Although heatmaps aim to mimic the optimal solution, the decoding process may produce a solution significantly different from the original imperfect heatmap in order to maintain feasibility. Since traditional SL-based heatmap solvers do not account for these effects during the decoding process, even if the generated heatmap is similar the optimal solution, the decoded solution may differ significantly, potentially leading to degraded performance.

### 3.3 DEPENDENCY ON TRAINING DATASET QUALITY

The final issue with existing SL-based heatmap solvers is their reliance on high-quality training datasets. These methods require large amounts of optimal solutions, but generating such datasets is computationally expensive due to the NP-hardness of most CO tasks. Current approaches rely on exact or sophisticated heuristic solvers to create training data, with time complexity increasing exponentially as the problem size grows, making this impractical, particularly for large-scale CO tasks. To address these challenges, enhancing model robustness using suboptimal datasets is crucial. These datasets are easier to generate, often requiring only brief heuristic runs.

## 4 METHOD

We propose an RL fine-tuning framework for diffusion models in CO. Our approach aims to address the described issues of SL-based heatmap solvers discussed in Section 3.

Figure 2: The overall framework of CADO

## 4.1 Markov Decision Process for Training Diffusion Models in CO

An MDP is defined by a tuple $(\mathcal{S}, \mathcal{A}, P, \rho_0, R)$, where $s \in \mathcal{S}$ is a state in the state space $\mathcal{S}$, $\mathbf{a} \in \mathcal{A}$ is an action belongs to the action space $\mathcal{A}$, $P(\mathbf{s}_{t+1} | \mathbf{s}_t, \mathbf{a}_t)$ is the state transition distribution, $\rho_0(\mathbf{s}_0)$ is the initial state distribution, and $R(\mathbf{s}_t, \mathbf{a}_t)$ is the reward function. The objective of RL is to learn a policy $\pi_\theta$ that maximizes the expected cumulative reward $J(\pi_\theta)$, formalized as $\mathbb{E}_{\tau \sim p(\tau | \pi_\theta)} \left[ \sum_{t=0}^{T} R(\mathbf{s}_t, \mathbf{a}_t) \right]$ where $\tau = (\mathbf{s}_0, \mathbf{a}_0 ... \mathbf{s}_T, \mathbf{a}_T)$ is a sequence of states and actions from a policy in the MDP.

Motivated from Black et al. (2024), we formulate the denoising process in the diffusion process as Markov Decision Process (MDP) for CO which is integrated the decoder $f_g$ algorithm during the training.

$$
\begin{aligned}
\mathbf{s}_t &\triangleq (g, t, \mathbf{x}_t), & \mathbf{a}_t &\triangleq \mathbf{x}_{t-1}, \\
\pi_\theta(\mathbf{a}_t | \mathbf{s}_t) &\triangleq p_\theta(\mathbf{x}_{t-1} | \mathbf{x}_t, g), & P(\mathbf{s}_{t+1} | \mathbf{s}_t, \mathbf{a}_t) &\triangleq (\delta_\mathbf{s}, \delta_{t-1}, \delta_{\mathbf{x}_{t-1}}), \\
\rho_0(\mathbf{s}_0) &\triangleq (g, t, \text{Bern}(\boldsymbol{p} = 0.5^{N_g})), & R(\mathbf{s}_t, \mathbf{a}_t) &\triangleq \begin{cases} -c_g(f_g(\mathbf{x}_0), g) & \text{if } t = 0, \\ 0 & \text{otherwise.} \end{cases}
\end{aligned}
\tag{8}
$$

where $\text{Bern}(\boldsymbol{p})$ is a Bernoulli distribution with vector probabilities $\mathbf{p}$ that samples the initial random noise $\mathbf{x}_T$, and $\delta_y$ is the Dirac delta distribution with nonzero density only at $y$. We then apply a policy gradient algorithm for optimizing the iterative denoising procedure with the cost function:

$$
\nabla_\theta J(\pi_\theta) = \mathbb{E} \left[ \sum_{t=0}^{T} \nabla_\theta \log p_\theta(\mathbf{x}_{t-1} | \mathbf{x}_t, g) \left( -c_g(f_g(\mathbf{x}_0), g) \right) \right].
\tag{9}
$$

If the heatmap-based solver can appropriately solve the MDP defined above, we can effectively address the existing issues in Section 3. Specifically, the generated heatmaps align with the true optimization objective, where the cost is calculated based on the decoded solution $f_g(\mathbf{x}_0, g)$ (Section 3.2), rather than merely imitating the optimal solutions (Section 3.1). Furthermore, during the RL fine-tuning process, the solver explores new solutions as actions, making the algorithm more robust when dealing with suboptimal solution datasets (Section 3.3).

T2T (Li et al., 2023), is another line of research that employs diffusion-based heatmap solver with cost incorporation. The key distinction between T2T and our approach lies in their treatment of cost information: while T2T considers costs during the inference phase by guiding the denoising process with the cost gradient, our method incorporates cost information during training through RL objective. A key advantage of T2T is that it avoids additional training overhead. However, T2T's performance is heavily dependent on the quality of the base diffusion model, and it deteriorates significantly if the underlying model is not well-trained. A comprehensive comparative analysis between these approaches is elaborated in Appendix F.

## 4.2 Training Process for Cost-Aware Diffusion Models

The whole process for training CADO is illustrated in Figure 2. CADO consists of two phases. In the first phase, the diffusion model is trained using the given dataset with the supervised learning objective $\mathcal{L}(\theta)$ in equation 7. In the second phase, we apply RL fine-tuning on the pretrained diffusion model to optimize $\mathcal{R}(\theta)$ in equation 9. To accurately measure the effectiveness of RL fine-tuning compared to previous works (Sun & Yang, 2023; Li et al., 2023), we directly finetune the pretrained

diffusion model called DIFUSCO (Sun & Yang, 2023). During this second phase, the training instances $g$ can be newly generated from the distribution $\boldsymbol{P}(g)$ or sampling from the instances in the train dataset.

In our RL fine-tuning process, we optimize a 12-layer GNN-based diffusion model using a hybrid approach. While the last one to two layers are completely unfrozen for full training, we apply Low-Rank Adaptation (LoRA) (Hu et al., 2022) to the remaining layers. Our techniques significantly improve both training stability and memory efficiency. Detailed explanations and comparative experimental results are elaborated in the Appendix C.2.

We use established simple greedy decoders to transform heatmaps into feasible solutions for Traveling Salesman Problem (TSP) and Maximum Independent Set (MIS) tasks, following the approaches of Sun & Yang (2023) and Li et al. (2023). For TSP, we include an optional post-processing step using the 2OPT heuristic (Lin & Kernighan, 1973) to refine solutions after decoding. We occasionally enhance our method with the local rewrite (LR) technique introduced by T2T (Li et al., 2023), which iteratively adds noise to disrupt the solutions and reconstructs sampled solutions to improve solution quality. However, unlike T2T, which leverages gradient-based cost guidance during local rewrite, our method does not utilize this guidance.

## 5 EXPERIMENT

The experiments are conducted using eight NVIDIA Tesla A40 GPUs for training and one Tesla A40 GPU for testing, along with two CPU cores of AMD EPYC 7413 24-Core Processor.

### 5.1 EXPERIMENT SETTINGS

**Problems.**  We test our proposed CADO on the Traveling Salesman Problem (TSP) and the Maximal Independent Set (MIS), which are basically edge and node selecting problems, respectively. TSP is the most commonly used benchmark combinatorial optimization problem, where the objective is to determine the shortest possible route that visits a set of nodes exactly once and returns to the original node. MIS is another widely used benchmark problem where the objective is to find the largest subset of vertices in a graph such that no two vertices in the subset are adjacent.

**Datasets.**  For RL fine-tuning, we generate new TSP instances for each problem size (TSP-50/100/500/1000/10000) by uniformly sampling the corresponding number of nodes from a unit square. We use identical test instances as Joshi et al. (2022); Kool et al. (2019b) for TSP-50/100 and Fu et al. (2021b) for TSP-500/1000/10000. Additionally, we evaluate our model on TSPLIB (Reinhelt, 2014), a real-world TSP benchmark dataset. For MIS experiments, we follow the dataset configurations used in previous studies (Li et al., 2018b; Ahn et al., 2020b; Böther et al., 2022; Qiu et al., 2022; Sun & Yang, 2023; Li et al., 2023), employing two graph types: SATLIB (Hoos & Stutzle, 2000) (MIS-SAT) and Erdős–Rényi (Erdos & Renyi, 1960) (MIS-ER). In contrast to TSP, we perform offline RL fine-tuning for MIS by reusing DIFUSCO's training instances rather than generating new, unseen instances.

**Evaluation Metrics.**  We assess our model and other baselines using three metrics. (1) Cost: For TSP, we measure the average tour length (lower is better). For MIS, we measure the average size of the independent set (higher is better). (2) Drop: We calculate the average performance difference between the model-generated solutions and optimal solutions. (3) Time: We record the total runtime during testing.

**Baselines.**  We compare our method with the following methods: (1) Exact Solvers: Concorde (Applegate et al., 2006) and Guruobi (Gurobi Optimization, 2020); (2) Heuristics : LKH3 (Applegate et al., 2006) and Farthest Insertion; (3) SL : GCN (Joshi et al., 2019a), BQ (Drakulic et al., 2023), LEHD (Luo et al., 2023), DIFUSCO (Sun & Yang, 2023), and T2T (Li et al., 2023); (4) RL : AM (Kool et al., 2019b), POMO (Hottung et al., 2021), DIMES (Qiu et al., 2022), ICAM (Zhou et al., 2024), GLOP (Ye et al., 2024) and UDC (Zheng et al., 2024).

We adapt T2T's experimental settings for fair comparison with our key baselines DIFUSCO and T2T. We focus on denoising steps in diffusion models, as increasing the number of diffusion steps

Table 1: Results on TSP-50 and TSP-100. AS: Active Search, S: Sampling Decoding, BS: Beam Search, RRC: Random Re-Construct (algorithm from Luo et al. (2023), which iteratively refines the partial solution). * represents the baseline for computing the drop. The results of models† are taken from each paper, and the rest of the results are taken from Li et al. (2023).

| Algorithm | Type | TSP-50 | | TSP-100 | |
|---|---|---|---|---|---|
| | | Length ↓ | Drop ↓ | Length ↓ | Drop ↓ |
| Concorde (Applegate et al., 2006) | Exact | 5.69* | 0.00% | 7.76* | 0.00% |
| 2OPT (Lin & Kernighan, 1973) | Heuristics | 5.86 | 2.95% | 8.03 | 3.54% |
| Farthest Insertion | Heuristics | 6.12 | 7.50% | 8.72 | 12.36% |
| AM (Kool et al., 2019b) | RL | 5.80 | 1.76% | 8.12 | 4.53% |
| GCN (Joshi et al., 2019a) | SL | 5.87 | 3.10% | 8.41 | 8.38% |
| Transformer (Bresson & Laurent, 2021) | RL | 5.71 | 0.31% | 7.88 | 1.42% |
| POMO (Kwon et al., 2020) | RL | 5.73 | 0.64% | 7.84 | 1.07% |
| Sym-NCO (Kim et al., 2022) | RL | - | - | 7.84 | 0.94% |
| Image Diffusion (Graikos et al., 2022b) | SL | 5.76 | 1.23% | 7.92 | 2.11% |
| BQ† (Drakulic et al., 2023) | SL | - | - | 7.79 | 0.35% |
| LEHD† (Luo et al., 2023) | SL | - | - | 7.81 | 0.58% |
| ICAM† (Zhou et al., 2024) | RL | - | - | 7.83 | 0.90% |
| DIFUSCO (Sun & Yang, 2023) | SL | 5.72 | 0.48% | 7.84 | 1.01% |
| T2T (Li et al., 2023) | SL | 5.69 | 0.04% | 7.77 | 0.18% |
| CADO (Ours) | SL+RL | **5.69** | **0.01%** | **7.77** | **0.08%** |
| AM (Kool et al., 2019b) | RL+2OPT | 5.77 | 1.41% | 8.02 | 3.32% |
| GCN (Joshi et al., 2019a) | SL+2OPT | 5.70 | 0.12% | 7.81 | 0.62% |
| Transformer (Bresson & Laurent, 2021) | RL+2OPT | 5.70 | 0.16% | 7.85 | 1.19% |
| POMO (Kwon et al., 2020) | RL+2OPT | 5.73 | 0.63% | 7.82 | 0.82% |
| Sym-NCO (Kim et al., 2022) | RL+2OPT | - | - | 7.82 | 0.76% |
| BQ† (Drakulic et al., 2023) | - | - | - | - | - |
| LEHD† (Luo et al., 2023) | SL+RRC | - | - | 7.76 | 0.01% |
| ICAM† (Zhou et al., 2024) | RL+RRC | - | - | 7.79 | 0.41% |
| DIFUSCO (Sun & Yang, 2023) | SL+2OPT | 5.69 | 0.09% | 7.78 | 0.22% |
| T2T (Li et al., 2023) | SL+2OPT | 5.69 | 0.02% | **7.76** | **0.06%** |
| CADO (Ours) | SL+RL+2OPT | **5.69** | **0.01%** | **7.76** | **0.06%** |

tends to improve performance but also results in longer inference times (Sun & Yang, 2023). In our experiments, DIFUSCO uses 120 steps for TSP-50/100 and 50 for other tasks. T2T and CADO use 50 steps for TSP-50/100 and 20 for others, plus Local Rewrite Search, matching DIFUSCO's resources. We also evaluated CADO-L, a lightweight version, which applies 20 diffusion steps across all tasks while eliminating additional heuristics such as Local Rewrite Search and 2OPT. This simplified version requires only 40% of the computational cost compared to the baselines. Note that the computational complexity during inference remains comparable across DIFUSCO, T2T, and CADO. However, the empirical differences observed in experiments stem from variations in GPU, PyTorch implementations, and optimization strategies.

## 5.2 MAIN RESULT

We divided the table into two parts for learning-based approaches. In Table 1, results are divided into two parts based on whether additional heuristics are used or not. In Table 2 and 3 the upper part shows the performance of models that generate a solution with just a single inference, while the lower part shows the performance of models that use multiple sampling and select the best solution among them. The experimental results on TSPLIB are in Appendix D.

**TSP-50/100.** Table 1 shows our TSP-50/100 results. The cost signals in training boosted performance to SOTA. On TSP-50 and TSP-100, our method demonstrates performance comparable to the state-of-the-art, regardless of whether 2OPT is used. This strongly validates the effectiveness of our approach.

**TSP-500/1000/10000.** For large-scale TSP-500/1000/10000 instances, CADO maintains stable and effective performance. As other baselines' performance degrades, our RL fine-tuning benefits become clearer. CADO outperforms other diffusion solvers across all criteria. Especially, without 2OPT, CADO achieves 1.54%, 4.42%, 10.73% in 500, 1000, 10000 respectively, which is a significant performance improvement over existing diffusion-based baselines DIFUSCO (9.41%, 11.24%, 36.75%) and T2T (6.92%, 9.83%, - %), indicating that our approach makes much better use of the

Table 2: Results on TSP-500/1000/10000. AS: Active Search, S: Sampling Decoding, BS: Beam Search, RRC: Random Re-Construct (Luo et al., 2023), which iteratively refines the partial solution. * represents the baseline for computing the drop. The results of models† are taken from each paper, and the rest of the results are taken from Li et al. (2023).

| Algorithm | Type | TSP-500 | | | TSP-1000 | | | TSP-10000 | | |
|---|---|---|---|---|---|---|---|---|---|---|
| | | Length ↓ | Drop ↓ | Time | Length ↓ | Drop ↓ | Time | Length ↓ | Drop ↓ | Time |
| Concorde (Applegate et al., 2006) | Exact | 16.55* | - | 37.66m | 23.12* | - | 6.65h | - | - | - |
| Gurobi (Gurobi Optimization, 2020) | Exact | 16.55 | 0.00% | 45.63h | - | - | - | - | - | - |
| LKH-3 (default) (Helsgaun, 2017) | Heuristics | 16.55 | 0.00% | 46.28m | 23.12 | 0.00% | 2.57h | 71.77* | - | 8.8h |
| Farthest Insertion | Heuristics | 18.30 | 10.57% | 0s | 25.72 | 11.25% | 0s | 80.59 | 12.29% | 6s |
| AM (Kool et al., 2019b) | RL | 20.02 | 20.99% | 1.51m | 31.15 | 34.75% | 3.18m | 141.68 | 97.39% | 5.99m |
| GCN (Joshi et al., 2019a) | SL | 29.72 | 79.61% | 6.67m | 48.62 | 110.29% | 28.52m | - | - | - |
| POMO+EAS-Emb (Hottung et al., 2021) | RL+AS | 19.24 | 16.25% | 12.80h | - | - | - | - | - | - |
| POMO+EAS-Tab (Hottung et al., 2021) | RL+AS | 24.54 | 48.22% | 11.61h | 49.56 | 114.36% | 63.45h | - | - | - |
| DIMES (Qiu et al., 2022) | RL | 18.93 | 14.38% | 0.97m | 26.58 | 14.97% | 2.08m | 86.44 | 20.44% | 4.65m |
| DIMES (Qiu et al., 2022) | RL+AS | 17.81 | 7.61% | 2.10h | 24.91 | 7.74% | 4.49h | 80.45 | 12.09% | 3.07h |
| DIMES (Qiu et al., 2022) | RL+2OPT | 17.65 | 6.62% | 1.01m | 24.83 | 7.38% | 2.29m | - | - | - |
| DIMES (Qiu et al., 2022) | RL+AS+2OPT | 17.31 | 4.57% | 2.10h | 24.33 | 5.22% | 4.49h | - | - | - |
| BQ† (Drakulic et al., 2023) | SL | 16.72 | 1.18% | 0.77m | 23.65 | 2.27% | 1.90m | - | - | - |
| LEHD† (Luo et al., 2023) | SL | 16.78 | 1.56% | 0.27m | 23.85 | 3.17% | 1.60m | - | - | - |
| LEHD† (Luo et al., 2023) | SL+RRC | 16.58 | 0.34% | 8.7h | 23.40 | 1.20% | 48.6m | - | - | - |
| ICAM† (Zhou et al., 2024) | RL | 16.78 | 1.56% | 0.02m | 23.80 | 2.93% | 0.03m | - | - | - |
| ICAM† (Zhou et al., 2024) | RL+RRC | 16.69 | 1.01% | 2.40m | 23.55 | 1.86% | 16.8m | - | - | - |
| GLOP† (Ye et al., 2024) | RL | 16.91 | 1.99% | 1.50m | 23.84 | 3.11% | 3.0m | 75.29 | 4.90% | 1.80m |
| UDC† (Zheng et al., 2024) | RL | 16.94 | 2.53% | 0.33m | 23.79 | 2.92 | 0.53m | 82.1 | 14.35% | 7.00m |
| DIFUSCO (Sun & Yang, 2023) | SL | 18.11 | 9.41% | 5.70m | 25.72 | 11.24% | 17.33m | 98.15 | 36.75% | 28.51m |
| DIFUSCO (Sun & Yang, 2023) | SL+2OPT | 16.81 | 1.55% | 5.75m | 23.55 | 1.86% | 17.52m | 73.99 | 3.10% | 35.38m |
| T2T (Li et al., 2023) | SL | 17.69 | 6.92% | 4.90m | 25.39 | 9.83% | 17.93m | - | - | - |
| T2T (Li et al., 2023) | SL+2OPT | 16.68 | 0.83% | 4.83m | 23.41 | 1.26% | 18.37m | - | - | - |
| **CADO (Ours)** | SL+RL | 16.80 | 1.54% | 3.74m | 24.14 | 4.42 % | 7.80m | - | - | - |
| **CADO (Ours)** | SL+RL+2OPT | **16.66** | **0.70%** | 3.78m | **23.32** | **0.88 %** | 8.34m | **73.72** | **2.72%** | 18.22m |
| EAN (Deudon et al., 2018) | RL+S+2OPT | 23.75 | 43.57% | 57.76m | 47.73 | 106.46% | 5.39h | - | - | - |
| AM (Kool et al., 2019b) | RL+BS | 19.53 | 18.03% | 21.99m | 29.90 | 29.23% | 1.64h | 129.40 | 80.28% | 1.81h |
| GCN (Joshi et al., 2019a) | SL+BS | 30.37 | 83.55% | 38.02m | 51.26 | 121.73% | 51.67h | - | - | - |
| DIMES (Qiu et al., 2022) | RL+S | 18.84 | 13.84% | 1.06m | 26.36 | 14.01% | 2.38m | 85.75 | 19.48% | 4.80m |
| DIMES (Qiu et al., 2022) | RL+AS+S | 17.80 | 7.55% | 2.11h | 24.89 | 7.70% | 4.53h | 80.42 | 12.05% | 3.12h |
| DIMES (Qiu et al., 2022) | RL+S+2OPT | 17.64 | 6.56% | 1.10m | 24.81 | 7.29% | 2.86m | - | - | - |
| DIMES (Qiu et al., 2022) | RL+AS+S+2OPT | 17.29 | 4.48% | 2.11h | 24.32 | 5.17% | 4.53h | - | - | - |
| BQ† (Drakulic et al., 2023) | SL+BS | 16.62 | 0.58% | 11.9m | 23.43 | 1.36% | 29.4m | - | - | - |
| ICAM† (Zhou et al., 2024) | RL+BS | 16.69 | 1.01% | 1.50m | 23.54 | 1.83% | 10.50m | - | - | - |
| ICAM† (Zhou et al., 2024) | RL+S | 16.65 | 0.78% | 0.63m | 23.49 | 1.58% | 3.80m | - | - | - |
| UDC† (Zheng et al., 2024) | RL + S | 16.78 | 1.58% | 4.00m | 23.53 | 1.78 | 8.00m | - | - | - |
| DIFUSCO (Sun & Yang, 2023) | SL+S | 17.48 | 5.65% | 19.02m | 25.11 | 8.61% | 59.18m | 95.52 | 33.09% | 6.72h |
| DIFUSCO (Sun & Yang, 2023) | SL+S +2OPT | 16.69 | 0.83% | 19.05m | 23.42 | 1.30% | 59.53m | 73.89 | 2.95% | 6.59h |
| T2T (Li et al., 2023) | SL+S | 17.14 | 3.60% | 17.05m | 24.85 | 7.51% | 1.12h | - | - | - |
| T2T (Li et al., 2023) | SL+S +2OPT | 16.62 | 0.46% | 17.02m | 23.31 | 0.85% | 1.17h | - | - | - |
| **CADO (Ours)** | SL+RL+S | 16.75 | 1.27% | 11.13m | 23.82 | 3.03 % | 26.8m | 78.8597 | 9.87% | 21.62m |
| **CADO (Ours)** | SL+RL+S+2OPT | **16.62** | **0.43%** | 11.17m | **23.26** | **0.63%** | 27.5m | **73.63** | **2.59%** | 48.72m |

Table 3: Results on SATLIB and ER-[700-800]

| Algorithm | Type | SATLIB | | | ER-[700-800] | | |
|---|---|---|---|---|---|---|---|
| | | Size ↑ | Drop ↓ | Time | Size ↑ | Drop ↓ | Time |
| KaMIS (Lamm et al., 2016) | Heuristics | 425.96* | - | 37.58m | 44.87* | - | 52.13m |
| Gurobi (Gurobi Optimization, 2020) | Exact | 425.95 | 0.00% | 26.00m | 41.28 | 7.78% | 50.00m |
| Intel (Li et al., 2018a) | SL | 420.66 | 1.48% | 23.05m | 34.86 | 22.31% | 6.06m |
| DIMES (Qiu et al., 2022) | RL | 421.24 | 1.11% | 24.17m | 38.24 | 14.78% | 6.12m |
| UDC† (Zheng et al., 2024) | RL | - | - | - | 41.00 | 8.62% | 0.67m |
| DIFUSCO (Sun & Yang, 2023) | SL | 424.56 | 0.33% | 8.25m | 36.55 | 18.53% | 8.82m |
| T2T (Li et al., 2023) | SL | **425.02** | **0.22%** | 8.12m | 39.56 | 11.83% | 8.53m |
| **CADO (Ours)** | SL+RL | 425.00 | 0.22% | 6.87m | **43.32** | **3.45%** | 4.28m |
| Intel (Li et al., 2018a) | SL+TS | - | - | - | 38.80 | 13.43% | 20.00m |
| DGL (Böther et al., 2022) | SL+TS | - | - | - | 37.26 | 16.96% | 22.71m |
| LwD (Ahn et al., 2020a) | RL+S | 422.22 | 0.88% | 18.83m | 41.17 | 8.25% | 6.33m |
| GFlowNets (Zhang et al., 2023) | UL+S | 423.54 | 0.57% | 23.22m | 41.14 | 8.53% | 2.92m |
| UDC† (Zheng et al., 2024) | RL+S | - | - | - | 42.88 | 4.44% | 21.05m |
| DIFUSCO (Sun & Yang, 2023) | SL+S | 425.13 | 0.19% | 26.32m | 40.35 | 10.07% | 32.98m |
| T2T (Li et al., 2023) | SL+S | **425.22** | **0.17%** | 23.80m | 41.37 | 7.81% | 29.73m |
| **CADO (Ours)** | SL+RL+S | 425.22 | 0.17% | 21.98m | **43.82** | **2.35%** | 13.53m |

diffusion model itself for CO. With 2OPT, CADO again achieves the best results among all NCO solvers in the table.

**MIS-SAT/ER.** For MIS problems, our approach shows promising results even with offline RL fine-tuning without generating new instances. In MIS-SAT, the performance improvement is minimal due to DIFUSCO's (0.33%) already saturated performance. In MIS-ER, despite the offline RL fine-tuning without exposure to unseen instances, we observe substantial improvements. These re-

Table 4: Analysis for effects of decoder.

| Algorithm | Test (Grdy) Drop ↓ | Test (NN) Drop ↓ |
|---|---|---|
| DIFUSCO | 1.62% | 2.32% |
| CADO-L (Grdy) | **0.27%** | 1.83% |
| CADO-L (NN) | 0.28% | **0.54%** |

Table 5: Results on the low quality dataset.

| Algorithm | w/o 2OPT Drop ↓ | w/ 2OPT Drop ↓ |
|---|---|---|
| DIFUSCO | 11.84% | 1.99% |
| T2T | 6.99% | 0.98% |
| CADO | **0.27%** | **0.08%** |

sults again strongly support our motivations discussed in Section 3, emphasizing the importance of learning that considers the effect of the decoding strategy and cost information rather than simply imitating optimal solutions, as done in traditional SL-based heatmap solvers.

## 5.3 EXPERIMENTAL VALIDATION: ADDRESSING ISSUES IN SL-BASED APPROACHES

We experimentally validate whether CADO really overcomes the issues of SL-based heatmap solvers outlined in Section 3 through systematic analysis and additional comparisons to other baseline heatmap solvers that utilize cost information. To clarify the performances of neural solvers, most of the results are recorded without the help of an additional 2OPT heuristic in TSP.

### 5.3.1 LESS SIMILAR HEATMAP, BETTER SOLUTION COST

To investigate whether heatmaps similar to optimal solutions necessarily are decoded into better solutions (Section 3.1), we examine CADO-L's learning curve from DIFUSCO (Epoch 0) during RL-finetuning to CADO-L (Epoch 3000) in Figure 3. We measure the similarity between the generated heatmaps and optimal solutions using KL loss, while evaluating the quality of the solution through drop. As training progressed, the drop (cost) improves significantly from 1.6% to 0.2%. Interestingly, despite our objective being solely cost minimization, the KL loss indeed increases. These results demonstrate that simply mimicking optimal solutions can be counterproductive in combinatorial optimization (Section 3), simultaneously validating that our proposed method successfully overcomes the issue in Section 3.1.

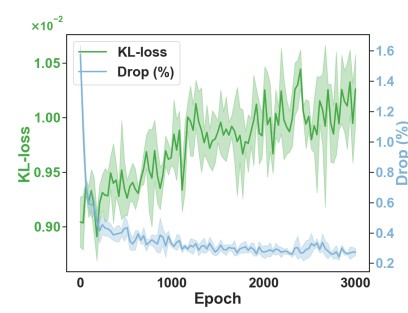

Figure 3: Learning curve of CADO-L.

### 5.3.2 THE EFFECT OF INCLUDING THE DECODER DURING TRAINING

To validate the decoder-related issues discussed (Section 3.2), we experiment with two simple decoders on TSP-100: **Grdy**, the standard decoder used in base CADO, and **NN**, which selects random initial nodes and moves to neighbors with the highest heatmap scores. We train two CADO variants, each with one of these decoders. All solvers are evaluated on both decoders using 20 inference steps without additional search techniques (2OPT/LR). The results in Table 4 strongly support our hypothesis (Section 3.2) that the choice of training decoder can significantly impact performance, even with identical heatmaps. The results show that each variant performed best with its corresponding training decoder: CADO-L (Grdy) achieving 0.27% with Test (Grdy) and CADO-L (NN) achieving 0.54% with Test (NN). Notably, for Test (NN), CADO-L (NN) significantly outperforms both CADO-L (Grdy) and DIFUSCO. These results demonstrate CADO's practical effectiveness in the decoder-related alignments.

### 5.3.3 TRAINING UNDER THE LOW QUALITY TRAIN DATASET

To investigate the impact of suboptimal training data on performance (Section 3.3), we evaluate models using a TSP-100 dataset containing 1.36% suboptimal solutions from LKH with a 1-second time limit. DIFUSCO shows significant degradation (11.84% w/o 2OPT, 1.99% w/ 2OPT), and while T2T's cost-guided search shows some improvement (6.99% w/o 2OPT, 0.98% w 2OPT), it still fall short. However, CADO maintains robust performance (0.27% w/o 2OPT, 0.08% w/ 2OPT) through

Table 6: Comparisons of heatmap-based solvers with different cost information integration strategies. **GS**: Gradient Search, **LR**: Local Rewrite. See Section 4.2 for more details.

| Algorithm | Type | | TSP-500 | | TSP-1000 | | SATLIB | | ER-[700-800] | |
| --- | --- | --- | --- | --- | --- | --- | --- | --- | --- | --- |
| | Train | Inference | Drop $\downarrow$ | Time | Drop $\downarrow$ | Time | Drop $\downarrow$ | Time | Drop $\downarrow$ | Time |
| DIMES (Qiu et al., 2022) | RL | - | 18.93% | 0.97m | 14.97% | 2.08m | 1.11% | 24.17m | 14.78% | 6.12m |
| DIFUSCO (Sun & Yang, 2023) | SL | - | 9.41% | 5.70m | 11.24% | 17.33m | 0.33% | 8.25m | 18.53% | 8.82m |
| T2T (Li et al., 2023) | SL | GS + LR | 6.92% | 4.90m | 9.83% | 17.93m | **0.22%** | 8.12m | 11.83% | 8.53m |
| CADO-L | SL + RL | - | **3.34%** | 1.43m | **6.70%** | 2.75m | 0.36% | 2.63m | **4.40%** | 1.60m |

RL fine-tuning, demonstrating that CADO effectively mitigates the instability of heatmap solvers on lower-quality datasets.

### 5.3.4 COMPARISONS TO OTHER COST INTEGRATED METHODS

In Table 6, we focus on how the utilization of solution datasets and cost information affects the performance of various heatmap-based CO solvers. The experimental results reveal that DIMES and DIFUSCO, which do not utilize either solution datasets or cost information, generally underperform compared to T2T and CADO-L, which leverage both components. When comparing T2T and CADO-L, CADO-L outperforms T2T with fewer inference steps on TSP-500/1000 and MIS-ER tasks, while T2T excels on MIS-SAT tasks where DIFUSCO also performs strongly. Our analysis suggests that CADO-L's fine-tuning approach can be more effective than T2T's cost-guided search without additional training, particularly when DIFUSCO's base performance is insufficient.

## 6 RELATED WORK

ML-based CO solvers can be categorized into autoregressive and heatmap-based solvers. Autoregressive solvers iteratively extend a partial solution until completion (Kool et al., 2019b; Bello et al., 2016; Kwon et al., 2020; Kim et al., 2022; Dernedde et al., 2024), but they struggle with scalability due to their sequential nature. SL-based heatmap solvers (Fu et al., 2021a; Geisler et al., 2022; Joshi et al., 2019a; Nowak et al., 2018) generate solutions in a single step, offering better scalability but often producing suboptimal solutions due to ignoring the post-process decoder and cost information during training. There is RL-based heatmap solvers (Qiu et al., 2022) in the literature but fails to perform well on large-scale problems. Recently, the divide and conquer framework has been used with both solver types to address large-scale problems by breaking them into smaller subproblems (Ye et al., 2024; Zheng et al., 2024).

Generative models, known for their success in image and text generation, have been adapted to CO for their expressive power (Graikos et al., 2022a; Mirhoseini et al., 2021; Kool et al., 2019a; Niu et al., 2020; Sun & Yang, 2023; Li et al., 2023). DIFUSCO, a diffusion model-based solver, shows promise in various CO problems (Sun & Yang, 2023). However, most generative models in CO rely on imitation learning, inheriting the same issues as heatmap-based solvers. Sanokowski et al. (2024) uses an unsupervised learning approach to directly optimize the cost function. Li et al. (2023), which is closely related to our work, extends DIFUSCO by integrating a cost-guided local search during the denoising process in inference, whereas our method incorporates cost information during training instead. One of the main strengths of T2T is its ability to bypass additional training overhead. However, its performance is highly reliant on the quality of the base diffusion model and can decline substantially if the underlying model is poorly trained.

## 7 CONCLUSION

In this paper, we introduced an RL fine-tuning framework for heatmap-based solvers, especially for diffusion model-based, that successfully addresses key issues from ignoring cost information in existing SL-based heatmap solvers, such as the disconnect between prediction quality and solution cost, as well as inefficiencies arising from excluding the post-process decoder during training. By integrating cost-based feedback and aligning the learning process with the final solution generation, our approach not only enhances model performance across various CO benchmarks but also improves scalability and efficiency, making it a promising advancement in neural CO.

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

# A   TRAINING OBJECTIVE IN DIFFUSION MODEL

Sun & Yang (2023) propose a diffusion model-based CO solver called DIFUSCO. In CO, a diffusion model is employed to estimate the distribution of high-quality solutions for combinatorial optimization problems during the training phase (Sun & Yang, 2023; Li et al., 2023). Since the solution $x$ is belongs to the discrete solution space $\{0,1\}^N$, the noising process $q(\mathbf{x_t}|\mathbf{x_{t-1}})$ and denoising process $q(\mathbf{x}_{t-1}|\mathbf{x}_t, \mathbf{x}_0)$ are also done on the discrete space $\{0,1\}^N$. In this work, we followed the discrete diffusion models introduced by Austin et al. (2021a); Hoogeboom et al. (2021); Sun & Yang (2023).

The diffusion process consists of a forward noising procedure and a reverse denoising procedure. The forward process incrementally adds noise to the initial solution $\mathbf{x_0} = x_\star^g$, creating a sequence of latent variables $\mathbf{x_0}, \mathbf{x_1}, \ldots, \mathbf{x_T}$. Note that in CO, $\mathbf{x_0}$ follows the high-quality solutions for a given instance $g$, i.e., $\mathbf{x_0} \sim \boldsymbol{P}(x_\star^g|g)$. Furthermore, the fully noised solution $\mathbf{x}_T$ in the last timestep $T$ becomes an $N_g$ dimensional Bernoulli random variable with probability $\mathbf{p} = \{0.5\}^{N_g}$ and each variable is independent of each other, i.e., $\mathbf{x}_T \sim \mathrm{Bern}(\mathbf{p} = \{0.5\}^{N_g})$. For brevity, we omit a problem instance $g$ and denote $x_\star^g$ as $\mathbf{x_0}$ in all formulas of the diffusion model as a convention.

The forward noising process is defined by $q(\mathbf{x_{1:T}}|\mathbf{x_0}) = \prod_{t=1}^T q(\mathbf{x_t}|\mathbf{x_{t-1}})$, where $\mathbf{x_0} \sim q(\mathbf{x_0}|g)$, and $q(\mathbf{x_{1:T}}|\mathbf{x_0}) = \prod_{t=1}^T q(\mathbf{x_t}|\mathbf{x_{t-1}})$ denotes the transition probability at each step. The reverse process is modeled as $p_\theta(\mathbf{x_{0:T}}|g) = p(\mathbf{x_T}) \prod_{t=1}^T p_\theta(\mathbf{x_{t-1}}|\mathbf{x_t}, g)$, with $\theta$ representing the model parameters. The training objective is to match $p_\theta(\mathbf{x_0}|g)$ with the data distribution $q(\mathbf{x_0}|g)$, optimized by minimizing the variational upper bound of the negative log-likelihood:

$$L(\theta) = \mathbb{E}_q \Big[ -\log p_\theta(\mathbf{x_0}|\mathbf{x_1}, g) + \sum_{t=2}^T D_{KL}(q(\mathbf{x_{t-1}}|\mathbf{x_t}, \mathbf{x_0}) \| p_\theta(\mathbf{x_{t-1}}|\mathbf{x_t}, g)) \Big] \tag{10}$$

In CO, considering that the entry of the optimization variable $x$ are indicators of whether to select a node or an edge, each entry can also be represented as an one-hot $\{0,1\}^2$ while modeling it with Bernoulli distribution. Therefore, for diffusion process, $x$ turns into $N$ one-hot vectors $\mathbf{x_0} \in \{0,1\}^{N \times 2}$. Then, discrete diffusion model Austin et al. (2021b) is utilized. Specifically, at each time step $t$, the process transitions from $\mathbf{x_{t-1}}$ to $\mathbf{x_t}$ defined as:

$$q(\mathbf{x_t}|\mathbf{x_{t-1}}) = \mathrm{Cat}(\mathbf{x_t}; \mathbf{p} = x_{t-1}\mathbf{Q_t}) \tag{11}$$

where the $\mathrm{Cat}(\boldsymbol{x}; \mathbf{p})$ is a categorical distribution over $x \in \{0,1\}^{N \times 2}$ with vector probabilities $\mathbf{p}$ and transition probability matrix $\mathbf{Q_t}$ is:

$$\mathbf{Q_t} = \begin{bmatrix} (1 - \beta_t) & \beta_t \\ \beta_t & (1 - \beta_t) \end{bmatrix} \tag{12}$$

Here, $\beta_t$ represents the noise level at time $t$. The t-step marginal distribution can be expressed as:

$$q(\mathbf{x_t}|\mathbf{x_0}) = \mathrm{Cat}(\mathbf{x_t}; \mathbf{p} = \mathbf{x_0}\overline{\mathbf{Q_t}}) \tag{13}$$

where $\overline{\mathbf{Q_t}} = \mathbf{Q_1}\mathbf{Q_2}, \ldots, \mathbf{Q}_t$. To obtain the distribution $q(\mathbf{x_{t-1}}|\mathbf{x_t}, \mathbf{x_0})$ for the reverse process, Bayes' theorem is applied, resulting in:

$$q(\mathbf{x_{t-1}}|\mathbf{x_t}, \mathbf{x_0}) = \mathrm{Cat}\left(\mathbf{x_{t-1}}; \mathbf{p} = \frac{\mathbf{x_t}\mathbf{Q_t}^\top \odot \mathbf{x_0}\overline{\mathbf{Q}}_{t-1}}{\mathbf{x_0}\overline{\mathbf{Q_t}}\mathbf{x_t}^\top}\right) \tag{14}$$

As in Austin et al. (2021b), the neural network responsible for denoising $p_\theta(\tilde{\mathbf{x}}_0|\mathbf{x_t}, g)$ is trained to predict the original data $\mathbf{x_0}$. During the reverse process, this predicted $\tilde{\mathbf{x}}_0$ is used as a substitute for $\mathbf{x_0}$ to calculate the posterior distribution:

$$p_\theta(\mathbf{x_{t-1}}|\mathbf{x_t}) = \sum_{\boldsymbol{x}} q(\mathbf{x_{t-1}}|\mathbf{x_t}, \tilde{\mathbf{x}}_0) p_\theta(\tilde{\mathbf{x}}_0|\mathbf{x_t}, g) \tag{15}$$

## B  Neural Network Architecture

Following Sun & Yang (2023), we also utilize an anisotropic graph neural network with edge gating Bresson & Laurent (2018a;b) for backbone network of the diffusion model.

Consider $h_i^\ell$ and $e_{ij}^\ell$ as the features of node $i$ and edge $ij$ at layer $\ell$, respectively. Additionally, let $t$ represent the sinusoidal features Vaswani et al. (2017) corresponding to the denoising timestep $t$. The propagation of features to the subsequent layer is performed using an anisotropic message-passing mechanism:

$$\hat{e}_{ij}^{\ell+1} = P^\ell e_{ij}^\ell + Q^\ell h_i^\ell + R^\ell h_j^\ell, \tag{16}$$

$$e_{ij}^{\ell+1} = e_{ij}^\ell + \mathrm{MLP}_e(\mathrm{BN}(\hat{e}_{ij}^{\ell+1})) + \mathrm{MLP}_t(t), \tag{17}$$

$$h_i^{\ell+1} = h_i^\ell + \alpha(\mathrm{BN}(U^\ell h_i^\ell + \sum_{j \in N_i} \sigma(\hat{e}_{ij}^{\ell+1}) \odot V^\ell h_j)), \tag{18}$$

where $U^\ell, V^\ell, P^\ell, Q^\ell, R^\ell \in \mathbb{R}^{d \times d}$ are learnable parameters for layer $\ell$, $\alpha$ denotes the ReLU activation function Krizhevsky et al. (2010), BN stands for Batch Normalization Ioffe & Szegedy (2015), $A$ signifies the aggregation function implemented as SUM pooling Xu et al. (2019), $\sigma$ is the sigmoid activation function, $\odot$ represents the Hadamard product, $N_i$ indicates the neighbors of node $i$, and $\mathrm{MLP}(\cdot)$ refers to a two-layer multi-layer perceptron.

For the Traveling Salesman Problem (TSP), the initial edge features $e_{ij}^0$ are derived from the corresponding values in $x_t$, and the initial node features $h_i^0$ are initialized using the nodes' sinusoidal features. In contrast, for the Maximum Independent Set (MIS) problem, $e_{ij}^0$ are initialized to zero, and $h_i^0$ are assigned values corresponding to $x_t$. We then apply a classification or regression head, with two neurons for classification and one neuron for regression, to the final embeddings of $x_t$ (i.e., $\{e_{ij}\}$ for edges and $\{h_i\}$ for nodes) for discrete and continuous diffusion models, respectively.

## C  Experiment Details

### C.1  Training Details for SL Learning

Since we leverage the trained checkpoints introduced by DIFUSCO (Sun & Yang, 2023) and T2T (Li et al., 2023), we adopt the datasets and training procedures mentioned in DIFUSCO. This approach ensures consistency with previous work and provides a solid foundation for our RL fine-tuning experiments.

| Training Details | TSP-50 | TSP-100 | TSP-500 | TSP-1000 | TSP-10000 | SATLIB | ER-[700-800] |
|---|---|---|---|---|---|---|---|
| Number of epochs | 50 | 50 | 50 | 50 | 50 | 50 | 50 |
| Number of instances | 1502000 | 1502000 | 128000 | 64000 | 6400 | 49500 | 163840 |
| Batch size | 512 | 256 | 64 | 64 | 8 | 128 | 32 |
| Learning rate schedule | Cosine schedule starting from 2e-4 and ending at 0 | | | | | | |
| Curriculum learning | No | No | Yes | Yes | Yes | No | No |
| Initialization | - | - | TSP-100 | TSP-100 | TSP-500 | - | - |

Table 7: DIFUSCO Training Details for different tasks

### C.2  Efficient RL Fine-tuning via LoRA and Selective Layer Training

RL fine-tuning of large models, such as diffusion models, typically suffers from training instability (Fan et al., 2023). To address this challenge and prevent reward hacking, we evaluate four parameter unfreezing strategies on our pre-trained 12-layer GNN model, using TSP-100 as our benchmark. All other hyperparameters remain constant in all experiments. Our baseline approach (**FULL**) unfreezes all parameters across the network. Although this offers maximum flexibility, it leads to significant training instability and memory inefficiency. We then explore more constrained approaches (**Last1** and **Last2**) by unfreezing only the final one or two layers of our GNN. These methods improve

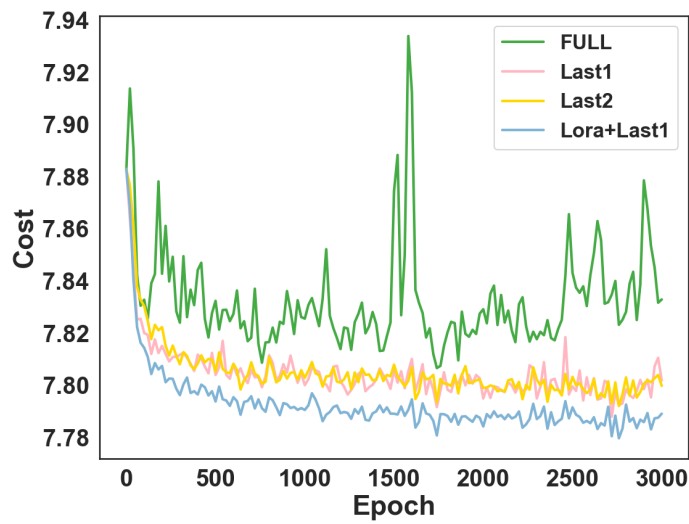

Figure 4: Learning curves for different training methods for RL-finetuning in TSP-100.

stability and memory efficiency, but show limited performance gains due to the high proportion of frozen parameters. To balance these trade-offs, we develop a hybrid approach (**Last1+LoRA**) that combines Last1 with Low-Rank Adaptation (**LoRA**). This method unfreezes the final layer while applying LoRA to the remaining layers. As shown in Figure 4, while FULL exhibits unstable training and suboptimal performance, and Last1/Last2 show minimal improvement despite their stability, our Last1+LoRA approach achieves both robust training and superior performance. Based on these experimental results, we adopt Last1+LoRA for training CADO across most benchmarks, with the exception of TSP-10000 only, where we employ Last2 for training efficiency.

### C.3 TRAINING DETAILS FOR RL-FINETUNING

Most hyperparameters remain consistent in all experiments, with the primary variation in the number of training epochs. For TSP-10000, we make two adjustments for training efficiency: we do not apply LoRA (Low-Rank Adaptation), and we increase the number of unfrozen layers in DIFUSCO (Last2). These modifications allow for more efficient training on this larger-scale problem while maintaining model performance.

| RL finetuning Details | TSP | | | | | MIS | |
|---|---|---|---|---|---|---|---|
| | 50 | 100 | 500 | 1000 | 10000 | SAT | ER |
| Number of epochs | 3000 | 3000 | 5000 | 5000 | 1250 | 3000 | 1400 |
| Number of samples in each epoch | 512 | | | | | | |
| Batch size | 64 | | | | | | |
| Learning rate | 1e-5 | | | | | | |
| Denoising step | 20 | 20 | 20 | 20 | 10 | 20 | 20 |
| LoRA Rank | 2 | 2 | 2 | 2 | 0 | 2 | 2 |
| Number of unfreezed Layers | 1 | 1 | 1 | 1 | 2 | 1 | 1 |

Table 8: RL finetuning Details for different tasks

## D    TSPLIB EXPERIMENT

**TSPLIB.**    To assess CADO's generalization ability, we tested the TSP100-trained model on TSPLIB instances with 50-200 nodes. CADO uses 2OPT and 4x sampling to solve these instances. Table 10 shows CADO outperforming other baselines again. It achieves a 0.117% performance,

Table 9: Comparison of Algorithm Performance on TSPLIB Instances. Results of other baselines are from Li et al. (2023); Hudson et al. (2021). 2OPT and sampling decoding are used in all diffusion-based models(DIFUSCO, T2T,CADO).

| Instances | AM | GCN | Learn2OPT | GNNGLS | DIFUSCO | T2T | CADO |
|---|---|---|---|---|---|---|---|
| eil151 | 16.767% | 40.025% | 1.725% | 1.529% | 0.314% | 0.314% | 0.000% |
| berlin52 | 4.169% | 33.225% | 0.449% | 0.142% | 0.000% | 0.000% | 0.000% |
| st70 | 1.737% | 24.785% | 0.040% | 0.764% | 0.172% | 0.000% | 0.000% |
| eil76 | 1.992% | 27.411% | 0.096% | 0.163% | 0.217% | 0.163% | 0.000% |
| pr76 | 0.816% | 27.793% | 1.228% | 0.039% | 0.043% | 0.039% | 0.000% |
| rat99 | 2.645% | 17.633% | 0.123% | 0.550% | 0.016% | 0.000% | 0.000% |
| kroA100 | 4.017% | 28.828% | 18.313% | 0.728% | 0.050% | 0.000% | 0.000% |
| kroB100 | 5.142% | 34.686% | 1.119% | 0.147% | 0.000% | 0.000% | 0.000% |
| kroC100 | 0.972% | 35.506% | 0.349% | 1.571% | 0.000% | 0.000% | 0.019% |
| kroD100 | 2.717% | 38.018% | 0.866% | 0.572% | 0.000% | 0.000% | 0.000% |
| kroE100 | 1.470% | 26.859% | 1.832% | 1.216% | 0.000% | 0.000% | 0.003% |
| rd100 | 3.407% | 50.432% | 1.725% | 0.003% | 0.000% | 0.000% | 0.000% |
| eil101 | 2.994% | 26.701% | 0.387% | 1.529% | 0.124% | 0.000% | 0.105% |
| lin105 | 1.739% | 34.902% | 1.867% | 0.606% | 0.441% | 0.393% | 0.450% |
| pr107 | 3.933% | 80.564% | 0.898% | 0.439% | 0.714% | 0.155% | 0.195% |
| pr124 | 3.677% | 70.146% | 10.322% | 0.755% | 0.997% | 0.584% | 0.340% |
| bier127 | 5.908% | 45.561% | 3.044% | 1.948% | 1.064% | 0.718% | 0.310% |
| ch130 | 3.182% | 39.090% | 0.709% | 3.519% | 0.077% | 0.077% | 0.019% |
| pr136 | 5.064% | 58.673% | 0.000% | 3.387% | 0.182% | 0.000% | 0.000% |
| pr144 | 7.641% | 55.837% | 1.526% | 3.581% | 1.816% | 0.000% | 0.222% |
| ch150 | 4.584% | 49.743% | 0.312% | 2.113% | 0.473% | 0.324% | 0.390% |
| kroA150 | 3.784% | 45.411% | 0.724% | 2.984% | 0.193% | 0.193% | 0.015% |
| kroB150 | 2.437% | 56.745% | 0.886% | 3.258% | 0.366% | 0.021% | 0.314% |
| pr152 | 7.494% | 33.925% | 0.029% | 3.119% | 0.687% | 0.687% | 0.806% |
| u159 | 7.551% | 38.338% | 0.054% | 1.020% | 0.000% | 0.000% | 0.001% |
| rat195 | 6.893% | 24.968% | 0.743% | 1.666% | 0.887% | 0.018% | 0.180% |
| d198 | 373.020% | 62.351% | 0.522% | 4.772% | 0.000% | 0.000% | 0.000% |
| kroA200 | 7.106% | 40.885% | 1.441% | 2.029% | 0.259% | 0.000% | 0.074% |
| kroB200 | 8.541% | 43.643% | 2.064% | 2.589% | 0.171% | 0.171% | 0.060% |
| Mean | 16.767% | 40.025% | 1.725% | 1.529% | 0.319% | **0.133%** | **0.117%** |

Table 10: Results on TSPLIB. Results of other baselines are from Li et al. (2023); Hudson et al. (2021)

| Algorithm | TSPLIB-[50-200] |
|---|---|
| | Drop ↓ |
| AM (Kool et al., 2019b) | 16.767% |
| GCN (Joshi et al., 2019a) | 40.025% |
| Learn2OPT (de O. da Costa et al., 2020) | 1.725% |
| GNNGLS (Hudson et al., 2021) | 1.529% |
| DIFUSCO (Sun & Yang, 2023) | 0.319% |
| T2T (Li et al., 2023) | 0.133% |
| **CADO (Ours)** | **0.117%** |

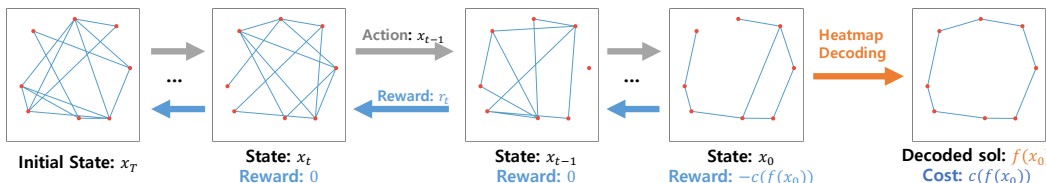

Figure 5: The overall denoise process in terms of MDP. The initial random noise $\mathbf{x}_T$ is sampled from the $\text{Bern}(\boldsymbol{p} = 0.5^N)$.

renewing approximately 13.6% over the previous best score. Detailed results of Table 10 are in Appendix D.

## E   Transfer Learning experiment

Table 11: Results on various TSP size.

| Fine-tuning | 100→500 Drop ↓ | 500→1000 Drop ↓ |
|---|---|---|
| SL → × | 3.2% | 2.12% |
| SL → SL (DIFUSCO) | 1.55% | 1.86% |
| SL → RL (CADO) | 1.59% | 1.04% |

In this section, we investigate transfer learning scenarios for scaled-up task sizes for TSP. We compare our primary RL fine-tuning approach with SL fine-tuning, which is feasible when a dataset for the target task is available, as in DIFUSCO. We examined two scenarios: (1) TSP100→TSP500 and (2) TSP500→TSP1000. Table 11 demonstrates that directly applying the model without fine-tuning results in poor performance. CADO (1.59%) achieves comparable performance to SL fine-tuning (1.55%) in the TSP100→TSP500 scenario and outperforms it in the TSP500→TSP1000 scenario (1.86% → 1.04%). Notably, CADO accomplishes this without requiring an additional dataset of optimal solutions for the target task sizes. These results highlight the effectiveness and efficiency of our RL fine-tuning approach in transfer learning settings, particularly for larger-scale problems.

## F   Comparison with Cost-aware Heatmap CO Solvers

Among heatmap-based CO solvers, several approaches also incorporate cost information with motivations similar to CADO. In this section, we highlight two important baselines: T2T (Li et al., 2023) and Dimes (Qiu et al., 2022), and compare them with CADO.

### F.1   Comparison with T2T

T2T (Li et al., 2023) is an important related work to our approach, as both employ diffusion-based heatmap CO solving with cost incorporation. The key distinction between T2T and our approach lies in their treatment of cost information: while T2T considers costs during inference through solution sampling without additional fine-tuning, CADO explicitly enable the model to sample low-cost solutions through supplementary training. Specifically, T2T's cost-guided sampling relies on two main concepts:

- **Cost-guided denoising process**: Similar to classifier guidance, this technique steers a well-trained diffusion model toward generating solutions with lower costs.
- **Local rewrite**: This diffusion-specific technique iteratively adds noise to disrupt the solutions and denoises the sampled solutions to obtain improved results.

By combining these approaches, T2T can generate low-cost solutions without additional fine-tuning. While this offers the advantage of avoiding CADO's extra training costs when the base heatmap

solver is well-trained, T2T's performance deteriorates if the underlying base diffusion model is inadequately trained. Our experimental results demonstrate these characteristics. As shown in Table 6, for MIS-SAT problems where the pretrained baseline DIFUSCO effectively learns optimal solution distributions, T2T outperforms CADO-L with its cost-guided sampling. Conversely, for TSP-500/1000 and MIS-ER problems where DIFUSCO may benefit from additional training refinements, CADO-L outperforms T2T with lower inference costs. Furthermore, our approach and T2T's method can be complementary. By incorporating T2T's Local rewrite technique (while excluding the cost-guided denoising process), CADO-L's performance improves substantially. In our main results, we refer to this hybrid approach simply as CADO.

### F.2 COMPARISON WITH DIMES

DIMES (Qiu et al., 2022) employs reinforcement learning to train the heatmap solver directly for high-quality solutions and naturally remains unaffected by the issues of SL-based heatmap solvers discussed in Section 3. Furthermore, DIMES offers an advantage over CADO by eliminating the need for additional solution datasets. However, this comes at the cost of being unable to leverage existing solution dataset information and the robust capabilities of diffusion models, resulting in somewhat inferior performance. To compensate, DIMES proposes instance-specific fine-tuning through local search or meta-learning frameworks during testing. While this approach significantly increases inference time, it still falls short of CADO in terms of solution quality generated from heatmaps.

