# OpenReview forum: "CADO: Cost-Aware Diffusion Models for Combinatorial Optimization via RL Fine-tuning"
_ICLR.cc/2025/Conference — Submitted to ICLR 2025_

### Official Review · Reviewer_waxg · 2024-10-21

**Soundness:** 3
**Presentation:** 3
**Contribution:** 3
**Rating:** 6
**Confidence:** 4

**Summary:**

This paper proposes CADO, a novel Cost-Aware Diffusion solver for combinatorial optimization (CO) tasks that address key limitations in traditional heatmap-based solvers, which typically rely on supervised learning to imitate optimal solutions but neglect cost information and the decoding process. By introducing a reinforcement learning (RL) fine-tuning phase, CADO integrates these elements directly into the training process, improving solution quality and computational efficiency. Applied to CO problems like the Traveling Salesman Problem (TSP) and Maximum Independent Set (MIS), CADO demonstrates superior performance, reducing computational costs by 15-40% and outperforming state-of-the-art solvers across various benchmarks.

**Strengths:**

1. This paper presents an interesting point that lower prediction errors do not always correlate with better solution quality. The introduction of an additional RL fine-tuning process is well-motivated to address the shortcomings of existing SL frameworks.

2. This paper introduces a novel two-phase training framework that combines SL with RL fine-tuning, addressing key limitations of traditional heatmap-based CO solvers.

3. The method design is reasonable, and the experimental results are convincing.

**Weaknesses:**

1. Cost guidance for models through post-training optimization has been explored in previous works, such as gradient search and local search [1,2]. The authors should address the differences between their approach and these methods, and discuss any potential advantages their method offers.

2. The methodology design appears somewhat simplistic, as it essentially applies RL on top of diffusion models. Yet, the strength of the paper, which includes motivation, discussion, and empirical results, helps to offset this limitation.

[1] T2T: From Distribution Learning in Training to Gradient Search in Testing for Combinatorial Optimization. NeurIPS 2023.

[2] DIMES: A Differentiable Meta Solver for Combinatorial Optimization Problems. NeurIPS 2022.

**Questions:**

1. What is the ratio of the training cost for RL fine-tuning compared to the training of the diffusion model?

2. Can the proposed method be combined with techniques such as gradient search [1] and local search [2], or were the empirical results already based on additional techniques during inference?

3. The paper uses the LoRA fine-tuning method. How would the results compare if full fine-tuning was used? What about the computational cost? And how do the results differ when using other fine-tuning methods?

4. Can this method be adapted to other supervised learning based models?

[1] T2T: From Distribution Learning in Training to Gradient Search in Testing for Combinatorial Optimization. NeurIPS 2023.

[2] DIMES: A Differentiable Meta Solver for Combinatorial Optimization Problems. NeurIPS 2022.

---

> ### Author Response · Authors · 2024-11-23
> **Author Response 1**
>
> We sincerely thank you for your insightful questions and constructive experimental suggestions that helped us better understand and expand our research directions.
>
> **[Comment 1]**
>
> Cost guidance for models through post-training optimization has been explored in previous works, such as gradient search and local search [1,2]. The authors should address the differences between their approach and these methods, and discuss any potential advantages their method offers.
>
> **[Response 1]**
>
> Thank you for raising this important point about algorithmic distinctions. We have provided a detailed comparison with baseline methods in Appendix F, which we can summarize as follows:
>
> First, regarding T2T [1], the key distinction lies in how cost information is utilized: T2T incorporates costs during inference by guiding the denoising process with cost gradients, while our method integrates cost information during training through reinforcement learning objectives. While T2T elegantly avoids additional training overhead, its performance significantly depends on the quality of the underlying diffusion model.
>
> Second, concerning DIMES [2], which uses direct reinforcement learning for heatmap solving without high-quality solutions, it exactly shares our advantage of being robust against the supervised learning-based solver issues discussed in Section 3. However, DIMES cannot leverage existing solution datasets or diffusion model capabilities, leading to lower performance. While DIMES attempts to compensate through instance-specific fine-tuning during testing, this approach increases inference time considerably while still not matching CADO's solution quality. In contrast, our method achieves superior performance without requiring any instance-specific search or fine-tuning at test time.
>
> **[Comment 2]**
>
> The methodology design appears somewhat simplistic, as it essentially applies RL on top of diffusion models. Yet, the strength of the paper, which includes motivation, discussion, and empirical results, helps to offset this limitation.
>
> **[Response 2]**
>
> We sincerely appreciate your thorough review that recognizes both the strengths of our work and provides constructive feedback.
>
> While you acknowledge the importance of our motivation, we want to emphasize that this crucial aspect has been largely overlooked in existing SL-based solvers [1,3,4,5,6,7], despite its significance. To further validate our motivations, we have conducted additional experiments in the revision that strongly demonstrate: (1) the existence of our identified problems and (2) the effectiveness of our RL-finetuning approach in addressing them.
>
> Recent studies have raised concerns about neural heatmap solver quality compared to rule-based heuristics [8], while others have shown promising results by incorporating cost information during inference through methods like MCTS [2,5,7,8] or Cost-guided search [1]. We believe our work clarifies these existing challenges in heatmap solvers and provides effective solutions, potentially influencing future research directions in the CO field.
>
> **[Comment 3]**
>
> What is the ratio of the training cost for RL fine-tuning compared to the training of the diffusion model?
>
> **[Response 3]**
>
> Thank you for raising this concern about computational efficiency. We acknowledge the importance of training costs in practical neural combinatorial optimization solvers. Our method requires relatively low computational overhead compared to training the base diffusion model from scratch through supervised learning. For instance, in the TSP-100 benchmark, our additional training requires only 15-20% of the time needed for the base model training.
>
> [1] Li, Yang, et al. "From distribution learning in training to gradient search in testing for combinatorial optimization." *Advances in Neural Information Processing Systems* 36 (2024).
>
> [2] Qiu, Ruizhong, Zhiqing Sun, and Yiming Yang. "Dimes: A differentiable meta solver for combinatorial optimization problems." *Advances in Neural Information Processing Systems* 35 (2022): 25531-25546.
>
> [3] Li, Zhuwen, Qifeng Chen, and Vladlen Koltun. "Combinatorial optimization with graph convolutional networks and guided tree search." Neruips 2018.
>
> [4] Joshi, Chaitanya K., Thomas Laurent, and Xavier Bresson. "An efficient graph convolutional network technique for the travelling salesman problem." Neurips 2019.
>
> [5] Fu, Zhang-Hua, Kai-Bin Qiu, and Hongyuan Zha. "Generalize a small pre-trained model to arbitrarily large tsp instances." AAAI 2021.
>
> [6] Geisler, Simon, et al. "Generalization of neural combinatorial solvers through the lens of adversarial robustness." **ICLR 2021.
>
> [7] Sun, Zhiqing, and Yiming Yang. "Difusco: Graph-based diffusion solvers for combinatorial optimization." Neurips 2023.
>
> [8] Xia, Yifan, et al. "Position: Rethinking Post-Hoc Search-Based Neural Approaches for Solving Large-Scale Traveling Salesman Problems." ICML 2024.

---

> ### Author Response · Authors · 2024-11-23
> **Authore Response 2**
>
> **[Comment 4]**
>
> Can the proposed method be combined with techniques such as gradient search [1] and local search [2], or were the empirical results already based on additional techniques during inference?
>
> **[Response 4]**
>
> We appreciate your insightful suggestions for extending our work.
>
> While incorporating local search into our framework could potentially enhance solution quality, our empirical analysis (DIMES [2], RL+AS in Table 2) indicates that this would introduce substantial computational overhead during inference. Given that our approach already learns highly accurate heatmaps, we believe that more efficient alternatives such as local rewrite or parallel sampling strategies would be more beneficial for practical applications.
>
> Regarding gradient search, while it could be naively integrated into the framework, our analysis suggests this may be detrimental to performance. Specifically, in CADO, when the RL fine-tuning process achieves optimal training, it inherently generates distributions that are precisely aligned with the corresponding decoder. In such cases, introducing additional cost guidance without careful consideration of the decoder dynamics could potentially compromise the performance rather than enhance it.
>
> It is worth to note that our method can be enhanced with the local rewrite (LR) technique introduced by T2T [1], which iteratively perturbs solutions through noise injection and reconstructs sampled solutions to improve solution quality. In our experimental setup, CADO-L represents our fine-tuned version without local rewrite, while CADO denotes the combined implementation of CADO-L and local rewrite.
>
> **[Comment 5]**
>
> The paper uses the LoRA fine-tuning method. How would the results compare if full fine-tuning was used? What about the computational cost? And how do the results differ when using other fine-tuning methods?
>
> **[Response 5]**
>
> We deeply appreciate your valuable suggestion that helps clarify one of our paper's key contributions. We have included additional experiments with various fine-tuning methods in the appendix C.2 of our revised manuscript.
>
> In the experiment, we address RL fine-tuning instability in our pre-trained 12-layer GNN model through four parameter unfreezing strategies: **FULL** (all parameters unfrozen), **Last1/Last2** (final one/two layers unfrozen), and **Last1+LoRA** (final layer unfrozen with Low-Rank Adaptation for remaining layers). While **FULL** suffers from instability and **Last1/Last2** show limited gains, **Last1+LoRA** (our adaptation) achieves both training stability and superior performance.
>
> Regarding computational efficiency, we quantify the relative training costs through the ratio of trainable parameters to total parameters for each method: **Last1 (8.3%), Last2 (16.6%), Last1+LoRA (9.7%), and FULL (100%)**. This analysis demonstrates that our proposed Last1+LoRA approach achieves optimal performance while maintaining computational efficiency, requiring only 9.7% of the parameters to be updated during fine-tuning.
>
> **[Comment 6]**
>
> Can this method be adapted to other supervised learning based models?
>
> **[Response 6]**
>
> Indeed, our proposed RL fine-tuning framework is model-agnostic and can be adapted to various supervised learning-based heatmap solvers. The framework enhances existing SL models by incorporating cost-awareness into the training process, and can be extended to any CO solver capable of generating intermediate outputs (e.g., heatmaps or partial solutions) suitable for reinforcement learning.
>
> Notably, we emphasize that our approach particularly benefits from two key advantages of diffusion models: their exceptional capability in learning high-dimensional heatmaps from an SL perspective, and their natural formulation of the denoising process as a multi-step MDP. We believe this strong synergy between diffusion models' inherent characteristics and our methodology significantly contributed to the superior performance of our approach.
>
> [1] Li, Yang, et al. "From distribution learning in training to gradient search in testing for combinatorial optimization." *Advances in Neural Information Processing Systems* 36 (2024).
>
> [2] Qiu, Ruizhong, Zhiqing Sun, and Yiming Yang. "Dimes: A differentiable meta solver for combinatorial optimization problems." *Advances in Neural Information Processing Systems* 35 (2022): 25531-25546.

---

> > ### Comment · Reviewer_waxg · 2024-11-28
> >
> > Thanks for the response. I have no further questions.

---

> > > ### Author Response · Authors · 2024-11-30
> > >
> > > We would like to express our sincere gratitude for your guidance and support throughout the review process. Your constructive feedback has been invaluable in helping us improve our work.

---

### Official Review · Reviewer_QGRc · 2024-10-31

**Soundness:** 3
**Presentation:** 3
**Contribution:** 3
**Rating:** 6
**Confidence:** 3

**Summary:**

This paper introduces RL finetuning to diffusion model-based combinatorial optimization solvers. The methodology seems to be built based on DIFUSCO and T2T, whereby a pretrained DIFUSCO is taken and fine-tuned with RL; and the decoding strategy in T2T is also considered. The main motivation and expected improvement is a CO solver should not only focus on mimicking the supervision signal but also consider the objective function to be minimized. Experiments on TSP and MIS show a little bit of marginal but solid improvements over state-of-the-arts.

**Strengths:**

* I found it interesting to involve objective function minimization in the training process. Introducing RL in training diffusion models for CO is a novel idea.
* The improvement over baselines such as T2T and DIFUSCO is solid among the benchmarks considered in this paper.

**Weaknesses:**

* The presentation does not discuss important peer methods properly, especially T2T. This paper is motivated by the fact that DIFUSCO does not consider the objective function and claims its contribution of involving the objective function by reinforcement learning. However, T2T has achieved consistent improvement over DIFUSCO by denoising with the guidance of the objective function (as one can tell from Tables 1,2,3), which the authors fully overlook. I understand the fact that this paper uses a different methodology (reinforcement learning), but T2T should be the most important peer method and the authors should not simply overlook that. Simply mentioning the following in related work is not enough
> (T2T and Sanokowski et al. (2024)) are similar to our work in that they utilized cost information, but those methods require a differentiable cost function
* Given the fact that 1) T2T is under a quite similar motivation, 2) T2T seems very competitive, the experimental improvement over T2T is less significant than the improvement over DIFUSCO, 3) the guided denoising strategy in T2T seems to be used in CADO (this paper)
> L304: We also enhance our method with an optional local search technique inspired by Chen & Tian (2019) and Li et al. (2023) called local rewriting. This involves strategically reintroducing noise followed by a secondary denoising process, allowing us to explore the solution space beyond initial denoising results

    I believe there needs more discussions and ablation studies on the introduced RL methodology: when comparing RL-finetuning (this paper) solely with guided denoising (T2T), which contributes more to the objective score? What is the understanding of faster inference on TSP but slower on MIS?
* Another important point this paper failed to address is its technical contribution: introducing RL to optimize the objective score is straightforward (there are tons of RL for CO related work on this), and fusing RL with diffusion is performed by following Black et al. (2024). What is the technical challenge resolved for the combination of RL, diffusion, and CO?

Typo/misc
* "Figure 5" is a table
* Citation for T2T in the upper half of Table 1 is wrong.

**Questions:**

I like the general idea of introducing RL to DIFUSCO and the solid improvement on TSP (better objective score and inference time) and improvement of objective score on MIS (slower inference, though). Before being fully convinced to suggest an acceptance for this paper, I would love to see more discussions on related work (especially T2T, which is missing in the current manuscript), and more insights among different ways of integrating the objective function into CO-solving.

I would like to suggest the following improvements in future versions & in the rebuttal:
* A dedicated subsection comparing CADO with T2T, focusing on these different approaches to incorporating cost information and their relative performance across various problem sizes and types.
* An ablation study that isolates the effects of RL fine-tuning and guided denoising. Additionally, A detailed analysis of the performance differences between TSP and MIS tasks will be nice, particularly focusing on the trade-offs between solution quality and inference time.
* A subsection to explicitly outline the novel technical challenges they encountered in combining RL, diffusion models, and CO. It will be nice to detail how your approach differs from simply applying existing techniques, and to highlight any innovations in your methodology that address CO-specific challenges.

One more technical question:
* This method is faster than T2T and DIFUSCO in Table 2. What is the methodology that made it possible? Why does it get slower than T2T and DIFUSCO in Table 3?

---

> ### Author Response · Authors · 2024-11-23
> **Authore Response 1**
>
> Thank you for your detailed feedback and for pointing out important aspects that have allowed us to further strengthen our research and presentation. We are especially pleased to see that you have a deep understanding of T2T, an important paper that shares a similar motivation with our work, and we sincerely appreciate your professional and insightful comments on this topic.
>
> **[Comment 1]**
>
> The presentation does not discuss important peer methods properly, especially T2T. This paper is motivated by the fact that DIFUSCO does not consider the objective function and claims its contribution of involving the objective function by reinforcement learning….
>
> **[Response 1]**
>
> We acknowledge the significance of T2T [1] as a crucial baseline and have added a detailed comparison in Section 4.1, with further comprehensive analysis in Appendix F. The fundamental difference between T2T and our approach lies in how cost information is utilized: T2T incorporates costs during inference by guiding the denoising process with cost gradients, while our method integrates cost information during training through an RL objective. While T2T benefits from avoiding additional training overhead, its performance critically depends on the quality of the underlying diffusion model and can degrade substantially with a suboptimal base model.
>
> **[Comment 2]**
>
> Given the fact that 1) T2T is under a quite similar motivation, 2) T2T seems very competitive, the experimental improvement over T2T is less significant than the improvement over DIFUSCO, 3) the guided denoising strategy in T2T seems to be used in CADO…
>
> **[Response 2]**
>
> We agree that comparing how the utilization of cost information affects experimental results is crucial. To address this, we restructure the table in Section 5.3.4 to focus on this comparison and provide detailed analyses and comparisons in Appendix F.1.
>
> We firstly need to clarify a potential misunderstanding. Our algorithm does not employ the cost-guided denoising proposed in T2T. While T2T's denoising phase comprises two stages—standard diffusion denoising followed by local rewriting—we only adopt the structure disruption aspect of local rewriting without implementing gradient-based cost guidance. This approach is sufficient since our heatmap solver is already fine-tuned with cost information. We have revised Section 4.2's conclusion to clearly articulate this distinction.
>
> As reviewer pointed out, direct comparison between RL-finetuning (CADO) solely and guided denoising (T2T) is also very important. To better highlight the effectiveness of our ablation study, we have created Table 6, which consolidates the key results from Tables 2 and 3 of our original submission, specifically comparing the performance of major baselines against CADO-L. In the table, we compare T2T with ablated version of CADO (CADO-L) which is a direct RL finetuned version of DIFUSCO [2] without using any techniques from T2T including local rewriting. We also comment that CADO-L requires less computation cost (about 40%) compared to other baselines: DIFUSCO and T2T.
>
> In Table 6, CADO-L outperforms T2T for TSP-500/1000 and MIS-ER, while T2T shows superior performance for MIS-SAT. T2T incorporates cost information during the post-inference stage, avoiding additional training costs, but consequently depends more heavily on the quality of the base diffusion model compared to CADO. Based on our analysis, this dependency explains the performance variations across different tasks. For MIS-SAT, where DIFUSCO achieves a highly accurate performance (0.33% error), additional RL fine-tuning becomes unnecessary, allowing T2T's cost-guided search to perform effectively. On the other hand, for tasks where DIFUSCO shows a relatively larger error (over 9%), CADO's fine-tuning proves to be more effective. Specifically, in the case of MIS-ER, where DIFUSCO exhibits an unstable performance with an 18.53% drop, the performance gap between T2T (11.83%) and CADO-L (4.40%) becomes the most pronounced.
>
> Finally, we believe T2T and CADO are complementary approaches to addressing SL-based heatmap solver limitations through cost information—T2T at the post-inference stage and CADO during training. Indeed, T2T's local rewrite technique significantly enhances CADO's performance. For instance, in TSP-1000, incorporating the local rewrite heuristic improves CADO-L's 6.70% gap to CADO’s 4.42%.
>
>
>
> [1] Li, Yang, et al. "From distribution learning in training to gradient search in testing for combinatorial optimization." *Advances in Neural Information Processing Systems* 36 (2024).
>
> [2] Sun, Zhiqing, and Yiming Yang. "Difusco: Graph-based diffusion solvers for combinatorial optimization." *Advances in Neural Information Processing Systems* 36 (2023): 3706-3731.
>
> [3] Qiu, Ruizhong, Zhiqing Sun, and Yiming Yang. "Dimes: A differentiable meta solver for combinatorial optimization problems." *Advances in Neural Information Processing Systems* 35 (2022): 25531-25546.

---

> ### Author Response · Authors · 2024-11-23
> **Authore Response 2**
>
> **[Comment 3]**
>
> Another important point this paper failed to address is its technical contribution: introducing RL to optimize the objective score is straightforward (there are tons of RL for CO related work on this), and fusing RL with diffusion is performed by following Black et al. (2024). What is the technical challenge resolved for the combination of RL, diffusion, and CO?
>
> **[Response 3]**
>
> Thank you for your insightful comment. We elaborate our contributions in [Common Response].
> While RL provides a natural framework for combinatorial optimization (CO) through direct cost optimization, its successful applications have been largely confined to autoregressive methods, where MDP formulation is straightforward and well-defined. Applying RL to heatmap solvers for large-scale CO problems presents significant challenges due to the high-dimensional nature of heatmap action spaces. Although DIMES [3] represents one of the few RL-based heatmap solvers capable of tackling large-scale CO problems like TSP, it still underperforms compared to SL-based approaches.
>
> To bridge this performance gap and enable effective direct RL training of heatmap solvers, we carefully design our algorithm to overcome issues existing in CO domains:
>
> - A two-stage learning approach that combines SL pretraining and RL fine-tuning, achieving both training stability and performance gains.
> - A diffusion-based framework that enables both refined supervised learning and well-defined MDP formulation through its denoising process. This dual-nature framework allows effective supervised pretraining while naturally reformulating CO tasks into multi-step MDPs, enabling robust scaling to large-scale problems.
> - We explored various fine-tuning techniques (details provided in Appendix C.2) and successfully stabilized the RL fine-tuning process in CO. This allowed for consistent training across tasks and hyperparameters, demonstrating the robustness, scalability, and applicability of our approach to future tasks
>
> To summarize, our work presents the first approach that integrates reinforcement learning, diffusion models, and combinatorial optimization with several practical innovations. This integration fundamentally addresses longstanding limitations of previous methods. Our comprehensive experiments validate the existence of these challenges in supervised learning-based heatmap solvers and demonstrate that our method effectively resolves them.
>
> **[Comment 4]**
>
> This method is faster than T2T and DIFUSCO in Table 2. What is the methodology that made it possible? Why does it get slower than T2T and DIFUSCO in Table 3?
>
> **[Response 4]**
>
> The computational complexity during inference is comparable between T2T and our method. While the fundamental operations remain similar, our implementation achieves faster practical performance through optimized code and updated Python dependencies, as detailed in Section 5.1 of the revised manuscript. Regarding the Maximum Independent Set (MIS) problem where our method showed slower performance in the initial submission, this was due to an inconsistency in batch sizes during evaluation—we had used batch size 1 while T2T used batch size 4. We have now updated our measurements using batch size 4 for a fair comparison. We appreciate the reviewer's attention to detail in identifying this discrepancy.
>
> [1] Li, Yang, et al. "From distribution learning in training to gradient search in testing for combinatorial optimization." *Advances in Neural Information Processing Systems* 36 (2024).
>
> [2] Sun, Zhiqing, and Yiming Yang. "Difusco: Graph-based diffusion solvers for combinatorial optimization." *Advances in Neural Information Processing Systems* 36 (2023): 3706-3731.
>
> [3] Qiu, Ruizhong, Zhiqing Sun, and Yiming Yang. "Dimes: A differentiable meta solver for combinatorial optimization problems." *Advances in Neural Information Processing Systems* 35 (2022): 25531-25546.

---

> > ### Comment · Reviewer_QGRc · 2024-11-25
> >
> > Thanks for the rebuttal. I have no further comments and will increase the score to 6

---

> > > ### Author Response · Authors · 2024-11-26
> > >
> > > We sincerely appreciate your recognition of CADO's framework and the improved score. Your insightful feedback, particularly regarding the other cost-aware solvers, has helped us significantly enhance the paper's comprehensiveness. We are once again grateful for your time and dedication in reviewing our work!

---

### Official Review · Reviewer_WKRP · 2024-11-01

**Soundness:** 2
**Presentation:** 3
**Contribution:** 2
**Rating:** 6
**Confidence:** 3

**Summary:**

In this paper, a cost-aware diffusion solver CADO for combinatorial optimization is proposed, and a fine-tuning framework based on reinforcement learning is proposed for diffusion models in combinatorial optimization. CADO consists of two phases: in the first phase, the diffusion model is trained using a given data set, and in the second phase, the pre-trained diffusion model is fine-tuned using reinforcement learning. Finally, it is applied to the traveling salesman and the maximal independent set problem.

**Strengths:**

1.The experiments are relatively full and complete, comparing multiple methods, and doing a wealth of experiments on different datasets for different problems.
2.Good results were obtained on different datasets.
3.To the best of our knowledge, current techniques using diffusion models as well as RL fine-tuning are not common in combinatorial optimization. This paper  provides a detailed code open source for the implementation of the algorithm, which provides a baseline for subsequent similar studies.

**Weaknesses:**

1.The innovation of the method is relatively low. The method of training and fine-tuning is very simple, which is slightly lacking as the innovation point of this paper. We do not see the authors' approach as a significant change from existing algorithms based on reinforcement learning fine-tuning.
2.The motivation also lists some basic problems in the field of combination optimization. It is not seen that these problems are essentially solved, especially (3.2) and (3.3). Alternatively, the authors do not do a good job of explaining why their approach is necessary to solve these difficult problems. In addition, (3.1) has always been a difficult problem faced by the traveling salesman problem, and there is no obvious groundbreaking.
3.Overall, after reading the entire article I'm still not sure why the algorithm in the article works. Why they achieved better results compared to other algorithms and why this framework is necessary for combinatorial optimization.

**Questions:**

We will consider revising the evaluation based on the answers to the following questions.
1.Please highlight the core contribution points of the article. The method of supervised learning and fine-tuning is more like a trick than sufficient to support the innovation of the article.
2.Specific method support for motivation. Taking feasible punishment and optimization goal as reinforcement learning reward is a very simple idea in the field of combinatorial optimization. Can this paper give more detailed and specific support for motivation.
3.We would like the authors to better explain why the framework in this paper is useful for the hard cases in combinatorial optimization. And what necessary adjustments the authors have made to the original reinforcement learning approach to ensure its applicability to combinatorial optimization problems. We recognize the validity of the experimental results in the article. But we also believe that a clear statement of the principles that make the algorithm effective is also necessary, especially for problems such as combinatorial optimization.

---

> ### Author Response · Authors · 2024-11-23
> **Author Response 1**
>
> We sincerely appreciate your thorough review and insightful comments, which have significantly contributed to improving our manuscript. As your questions address fundamental aspects of our work, we have provided detailed responses in [Common Response]. We encourage you to reference these alongside our specific responses below, as they provide complementary context for our discussion.
>
> **[Comment 1]**
>
> 1.The innovation of the method is relatively low. The method of training and fine-tuning is very simple, which is slightly lacking as the innovation point of this paper. We do not see the authors' approach as a significant change from existing algorithms based on reinforcement learning fine-tuning. 2.The motivation also .....
>
> **[Response 1]**
>
> We deeply appreciate the reviewers' insightful questions that address fundamental aspects of our paper.
>
> We acknowledge that our identified issues might appear basic rather than groundbreaking. However, our literature survey reveals that surprisingly, from early works to very recent developments, most SL-based heatmap solvers for CO problems have learned heatmaps through imitation without considering cost information [1,2,3,4,5,7]. While RL-based approaches effectively incorporate these elements, their application to large-scale TSP remains limited (e.g., DIMES [6]) due to the high-dimensional action space, generally yielding inferior performance compared to SL-based methods.
>
> Existing research has largely overlooked both our seemingly obvious motivation and the integration of cost information in heatmap learning. For instance, [4] briefly addresses their choice of binary loss over optimality gap, stating merely that "Optimality gap is nature choice but it is proved to be tough to backpropagate through the decoding of the final solution from the soft prediction," indicating limited consideration of this crucial issues.
>
> We hypothesize two main reasons for this trend in previous works:
>
> - Researchers might have assumed that with ideal heatmap learning perfectly reproducing optimal solutions, issues (3.1) and (3.2) would become negligible, while (3.3) would be practically non-existent.
> - The inherent difficulty of effectively incorporating cost information in large-scale CO problems through methods like RL may have discouraged deeper exploration.
>
> However, our novel experiments in Section 5.3 strongly suggest that previous works underestimated the CO domain's sensitivity to even minor heatmap errors. Our three carefully designed experiments demonstrate that SOTA SL-based heatmap solvers are indeed vulnerable to these issues, respectively. Importantly, our proposed simple RL-finetuning significantly mitigates these vulnerabilities. These experimental results validate our motivation and explain our superior performance across most benchmarks in Section 5.2.
>
> Recent studies have raised concerns about neural heatmap solver quality compared to rule-based heuristics [8], while others have shown promising results by incorporating cost information during inference through methods like MCTS [3,5,6,7] or Cost-guided search [7]. We believe our work clarifies these existing challenges in heatmap solvers and provides effective solutions, potentially influencing future research directions in the CO field.
>
> [1] Li, Zhuwen, Qifeng Chen, and Vladlen Koltun. "Combinatorial optimization with graph convolutional networks and guided tree search." Neruips 2018.
>
> [2] Joshi, Chaitanya K., Thomas Laurent, and Xavier Bresson. "An efficient graph convolutional network technique for the travelling salesman problem." Neurips 2019.
>
> [3] Fu, Zhang-Hua, Kai-Bin Qiu, and Hongyuan Zha. "Generalize a small pre-trained model to arbitrarily large tsp instances." AAAI 2021.
>
> [4] Geisler, Simon, et al. "Generalization of neural combinatorial solvers through the lens of adversarial robustness." **ICLR 2021.
>
> [5] Sun, Zhiqing, and Yiming Yang. "Difusco: Graph-based diffusion solvers for combinatorial optimization." Neurips 2023.
>
> [6] Qiu, Ruizhong, Zhiqing Sun, and Yiming Yang. "Dimes: A differentiable meta solver for combinatorial optimization problems." *Advances in Neural Information Processing Systems* 35 (2022): 25531-25546.
>
> [7] Li, Yang, et al. "From distribution learning in training to gradient search in testing for combinatorial optimization." *Advances in Neural Information Processing Systems* 36 (2024).
>
> [8] Xia, Yifan, et al. "Position: Rethinking Post-Hoc Search-Based Neural Approaches for Solving Large-Scale Traveling Salesman Problems." ICML 2024.

---

> ### Author Response · Authors · 2024-11-23
> **Author Response 2**
>
> **[Comment 2]**
>
> Please highlight the core contribution points of the article. The method of supervised learning and fine-tuning is more like a trick than sufficient to support the innovation of the article.
>
> **[Response 2]**
>
> We thank the reviewer for their feedback on algorithmic contributions. We have enhanced our manuscript with additional experiments and analyses, as detailed in [Common Response].
>
>  Our core contributions are:
>
> 1. We identify three fundamental limitations unique in existing SL-based heatmap CO solvers (cost ignorance, decoder ignorance, dataset dependency). We find this limitations can be overcomed by the reinforcement learning, and propose the first RL fine-tuning framework to overcome the limitations by RL.
> 2. We propose a novel framework that synergistically combines SL, RL, and diffusion models, leveraging their complementary strengths to solve CO. Our framework, enhanced with our training stabilization techniques, effectively addresses the unique challenges inherent in CO problems.
> 3. We design novel experimental protocols to empirically validate each of our identified limitations in existing SL-based models. Our results conclusively demonstrate both the presence of these limitations and the effectiveness of our proposed solutions.
>
>
>
> **[Comment 3]**
>
> Specific method support for motivation. Taking feasible punishment and optimization goal as reinforcement learning reward is a very simple idea in the field of combinatorial optimization. Can this paper give more detailed and specific support for motivation.
>
> **[Response 3]**
>
> We acknowledge the importance of the issue raised by the reviewer and believe that clarifying this aspect significantly enhances the quality of our paper. Accordingly, we include the following content in the revision.
>
> To overcome the inherent limitations of conventional SL-based heatmap solvers, we strategically adopt an MDP framework that integrates diffusion models with decoders. Section 4.1 provides a comprehensive explanation of how this RL-based formulation fundamentally addresses the shortcomings that previous SL approaches could not resolve.
>
> We present experimental results in Section 5.3 that validate our algorithms' effectiveness in addressing each identified challenge. The results confirm both the presence of highlighted risks in existing SL-based solvers and the efficacy of our proposed solutions.
>
> Specifically, the updated content in Section 5.3 includes the following points:
>
> 5.3.1 How KL loss and cost changes during RL fine-tuning.
>
> 5.3.2 How performance varies depending on whether decoders are included in the training process.
>
> 5.3.3 How performance varies depending on train dataset quality.
>
> 5.3.4 Comparison  with various cost utilizing algorithms.
>
> We believe these revisions and additions address the concerns raised and provide clearer and more detailed support for the identified limitations and our proposed solutions.
>
> **[Comment 4]**
>
> We would like the authors to better explain why the framework in this paper is useful for the hard cases in combinatorial optimization.
>
> **[Response 4]**
>
> In theory, if an SL-based solver could be trained perfectly (although it is almost impossible in real scenario), it would not suffer from the limitations discussed in Section 3. For relatively simple CO tasks such as TSP-50, TSP-100, or MIS-SAT, SL-based solvers can indeed achieve sophisticated performance levels, thus mitigating these limitations. However, as problem complexity increases (e.g., with larger TSP instances or MIS-ER), SL-based solvers tend to exhibit unstable training behavior, and the three limitations we identified become more pronounced. Our RL fine-tuning approach effectively addresses these scalability challenges, as demonstrated through comprehensive experiments in Section 5.3.
>
> **[Comment 5]**
>
> And what necessary adjustments the authors have made to the original reinforcement learning approach to ensure its applicability to combinatorial optimization problems.
>
> **[Response 5]**
>
> While reinforcement learning (RL) offers direct cost optimization for combinatorial optimization (CO), its application to large-scale CO problems has been limited, especially for heatmap solvers where RL approaches like DIMES underperform compared to supervised learning methods. To develop a practical and effective heatmap solver for large-scale CO problems, we introduce the following key methodological advances:
>
> - A two-stage learning approach that combines SL pretraining and RL fine-tuning, achieving both training stability and performance gains.
> - A diffusion-based framework that reformulates CO tasks into well-defined multi-step MDPs, enabling effective scaling to large-scale problems.
> - Novel training stability techniques, including Low Rank Adaptation and selective parameter freezing, to enhance RL fine-tuning reliability. Details of these experiments are provided in the appendix C.2.

---

### Official Review · Reviewer_k7qD · 2024-11-03

**Soundness:** 3
**Presentation:** 3
**Contribution:** 2
**Rating:** 5
**Confidence:** 3

**Summary:**

This paper proposes a new Cost-Aware Diffusion solver for combinatorial Optimization (CADO) method via RL finetuning. The existing approaches simply imitate the shape of the solution without considering the cost information. In addition, they do not account for the decoding process in the overall network. Lastly, they heavily rely on high-quality training datasets, which are expensive to collect. The proposed method trains a diffusion model with two phases: 1) SL and 2) RL fine-tuning. The proposed method incorporates the cost information and the decoder, which enhance the overall performance with significant efficacy.

**Strengths:**

- The existing three problems and motivations are well described.
- Experiments from small-scale to large-scale problems demonstrate the effectiveness of the proposed method.

**Weaknesses:**

- Figure 1 shows the cost compared to the loss. However, the value difference is really small, which is not impressive.
- In two-phase training, the first phase is the same as DIFUSCO, and the second phase is just adoption of the existing decoder and RL algorithm. Overall, the method is really simple, and lacks a strong sense of novelty.
- The paper claims that the decoded solution may differ significantly, leading to degraded performance of the solver. Are there any example figures or quantitative results? From the main experimental tables, the performance quality is not significantly degraded, as the paper described.
- From the tables, the performance gap is minimal, making the results less impressive. In addition, it is uncertain whether the cost is significantly lower compared to other methods. In a certain dataset, the proposed method rather underperformed, showing inconsistent results.
- From table 4, it is unclear how sensitive the method is to dataset quality compared to existing algorithms. Moreover, this low-quality dataset setting does not look like a realistic setting.

**Questions:**

- Is the cost really problem? This task (CO) does not require a real-time solution, which means the cost gap between the proposed method and existing methods is not a significant problem.
- The proposed method trains the network with one more stage compared to the existing method. Is it a fair comparison? I wonder why the proposed method requires less time consumption compared to DIFUSCO. Moreover, as the experimental result shows, when DIFUSCO trains one more stage with the same SL, the performance is enhanced. It means the proposed method needs to be compared with other algorithms that run the same number of stages or iterations.

---

> ### Author Response · Authors · 2024-11-23
> **Author Response 1**
>
> Thank you for taking the time to thoroughly review our work and providing detailed and constructive feedback. We greatly appreciate your insights and suggestions.
>
> **[Comment 1]**
>
> Figure 1 shows the cost compared to the loss. However, the value difference is really small, which is not impressive.
>
> **[Response 1]**
>
> We have relocated the original Figure 1 to Figure 3 in Section 5.3.1. We have revised the visualization as we modified the y-axis scale to show the performance drop rather than cost. The updated Figure 3 demonstrates that CADO achieves a significant reduction in performance drop from 1.6% (DIFUSCO [1]) to 0.2% - an eightfold improvement in the context of CO.
>
> **[Comment 2]**
>
> In two-phase training, the first phase is the same as DIFUSCO, and the second phase is just adoption of the existing decoder and RL algorithm. Overall, the method is really simple, and lacks a strong sense of novelty.
>
> **[Response 2]**
>
> We appreciate the reviewer's observation regarding the apparent simplicity of our method.
>
> Our framework's novelty lies in systematically identifying three critical limitations of existing supervised learning-based heatmap solvers that emerge from naive imitation of optimal solutions. As elaborated in our [Common Response], our key technical contributions include:
>
> 1. The first RL fine-tuning framework for heatmap-based CO solvers that directly addresses these limitations of previous SL-based heatmap solvers.
> 2. Novel training stability techniques, including Low Rank Adaptation and selective parameter freezing, that enable reliable RL fine-tuning at scale
> 3. A diffusion-based framework that excels as an SL-based heatmap solver while naturally providing a well-defined multi-step MDP through its denoising process, enabling effective RL application
> 4. Comprehensive design of a series of novel experiments to verify the existence of the aforementioned limitations
>
> We note that the simplicity of our method proves to be a significant advantage as a practical Neural CO solver. Our model achieves robust state-of-the-art performance across diverse CO benchmarks while maintaining consistent hyperparameters across problems, demonstrating both exceptional generalizability and practical applicability.
>
> **[Comment 3]**
>
> The paper claims that the decoded solution may differ significantly, leading to degraded performance of the solver. Are there any example figures or quantitative results? From the main experimental tables, the performance quality is not significantly degraded, as the paper described.
>
> **[Response 3]**
>
> We sincerely appreciate the reviewer's constructive feedback on improving our manuscript's clarity. To address these concerns, we have substantially strengthened our empirical validation in Section 5.3.2. Our new experimental results demonstrate that decoder selection significantly impacts solution quality (1.83% → 0.27%), even when using an identical heatmap solver.
>
> Furthermore, we conducted a quantitative analysis on TSP-1000 to measure edge modification rates during the decoding process. Our findings reveal that 6.5% of edges initially present in the heatmap disappear during the decoding step to construct feasible solutions. In the context of TSP, where solution quality is highly sensitive to edge configurations, these modifications can substantially influence the final solution quality.
>
> **[Comment 4]**
>
> From the tables, the performance gap is minimal, making the results less impressive. In addition, it is uncertain whether the cost is significantly lower compared to other methods. In a certain dataset, the proposed method rather underperformed, showing inconsistent results.
>
> **[Response 4]**
>
> When evaluating Combinatorial Optimization (CO) algorithms, the standard practice is to measure performance through drops (optimality gaps) rather than solution costs. Under this metric, our approach demonstrates superior performance compared to existing baselines across the benchmark suite, with the sole exception of MIS-SAT, where it still achieves SOTA. This performance advantage becomes particularly pronounced as we scale to more challenging problem instances, as evidenced in TSP-500/1000/1000 and MIS-ER.
>
> **[References]**
>
> [1] Sun, Zhiqing, and Yiming Yang. "Difusco: Graph-based diffusion solvers for combinatorial optimization." Neurips 2023.
>
> [2] Li, Yang, et al. "From distribution learning in training to gradient search in testing for combinatorial optimization." *Advances in Neural Information Processing Systems* 36 (2024).

---

> ### Author Response · Authors · 2024-11-23
> **Author Response 2**
>
> **[Comment 5]**
>
> From table 4, it is unclear how sensitive the method is to dataset quality compared to existing algorithms. Moreover, this low-quality dataset setting does not look like a realistic setting.
>
> **[Response 5]**
>
> We appreciate the reviewer's comments regarding the clarity of our explanations.
>
> First, we would like to note that Table 4 has been incorporated into Table 5 in the revised manuscript.
> The results in Table 5, where all models are trained on suboptimal datasets, demonstrate the robustness of our RL finetuning framework compared to other heatmap solvers. In the w/o 2opt setting, our model achieves substantial improvements, reducing the drop from 11.84% to 0.27% (97.92% improvement) compared to DIFUSCO [1], and from 6.99% to 0.27% (96.14% improvement) compared to T2T [2]. These improvements persist even with 2OPT heuristic applied (w/ 2OPT), where our model reduces the drop from 1.99% to 0.08% (95.98% improvement) versus DIFUSCO, and from 0.98% to 0.08% (91.84% improvement) versus T2T [2].
>
> We emphasize that relying solely on optimal solutions is impractical for real-world applications, given the NP-hard nature of CO problems makes generating large-scale optimal datasets infeasible. For instance, the widely-used TSP-10,000 dataset contains only 6,400 training samples, which are suboptimal solutions generated by LKH3 within a time limit rather than optimal ones. Additionally, real-world data created by human experts often proves to be suboptimal.
>
> In conclusion, we believe our proposed setting reflects realistic scenarios, and our method's demonstrated robustness under these conditions directly addresses real-world application needs. This underscores the importance of developing algorithms that maintain strong performance when trained on suboptimal datasets.
>
> **[Comment 6]**
> Is the cost really problem? This task (CO) does not require a real-time solution, which means the cost gap between the proposed method and existing methods is not a significant problem.
>
> **[Response 6]**
>
> Although Combinatorial Optimization (CO) tasks may not always demand real-time solutions, computational efficiency remains a critical practical consideration. As demonstrated in Table 2, traditional exact solvers like LKH exhibit substantial computational requirements (8.8 hours for TSP-10,000), with execution times growing exponentially with problem size. Our method offers an attractive trade-off by delivering near-optimal solutions significantly faster, making it well-suited for real-world applications where computational efficiency is essential.
>
> **[Comment 7]**
>
> I wonder why the proposed method requires less time consumption compared to DIFUSCO.
>
> **[Response 7]**
>
> We appreciate the reviewer's astute observation regarding computational efficiency.
> While the theoretical inference time complexity of DIFUSCO [1] and CADO is indeed similar, our experimental results show that CADO achieves faster practical inference speeds. This performance advantage stems from our optimized implementation and utilization of updated Python libraries that enhance execution efficiency. We have included this clarification in Section 5.1 of the revised manuscript.
>
> **[References]**
>
> [1] Sun, Zhiqing, and Yiming Yang. "Difusco: Graph-based diffusion solvers for combinatorial optimization." Neurips 2023.
>
> [2] Li, Yang, et al. "From distribution learning in training to gradient search in testing for combinatorial optimization." *Advances in Neural Information Processing Systems* 36 (2024).

---

> ### Author Response · Authors · 2024-11-23
> **Author Response 3**
>
> **[Comment 8]**
>
> The proposed method trains the network with one more stage compared to the existing method. Is it a fair comparison? Moreover, as the experimental result shows, when DIFUSCO trains one more stage with the same SL, the performance is enhanced. It means the proposed method needs to be compared with other algorithms that run the same number of stages or iterations.
>
> **[Response 8]**
>
> We appreciate the reviewer's concerns regarding the fairness of our additional training stage. We acknowledge that training computational costs are a significant consideration in developing practical algorithms.
>
> We would like to address several key points:
> First, our RL fine-tuning stage requires relatively modest computational resources compared to the initial SL training. Specifically, in our TSP-100 experiments, CADO incurs only 15%-20% additional training costs compared to our base SL-based heatmap solver (DIFUSCO [1]), while achieving substantial performance improvements.
>
> Second, it's important to note that DIFUSCO was trained to convergence - additional SL training would not necessarily yield further improvements. In fact, we observe that the converged diffusion model encounters fundamental limitations inherent to the SL-based heatmap solver's objective. This is evidenced in Figure 3, where the SL objective loss increases despite improving actual performance.
>
> Regarding your observation about DIFUSCO's performance enhancement with additional SL training: This refers to the transfer learning experiments presented in Table 11 of our revised manuscript. The improved performance in the SL → SL setup (compared to SL → x) stems from successfully SL fine-tuning DIFUSCO from TSP-100 to previously unseen TSP-500 instances, rather than indicating a general performance boost from additional training.
>
> We trust these clarifications address your concerns about the fairness and efficiency of our approach.
>
> **[References]**
>
> [1] Sun, Zhiqing, and Yiming Yang. "Difusco: Graph-based diffusion solvers for combinatorial optimization." Neurips 2023.
>
> [2] Li, Yang, et al. "From distribution learning in training to gradient search in testing for combinatorial optimization." *Advances in Neural Information Processing Systems* 36 (2024).

---

> > ### Comment · Reviewer_k7qD · 2024-11-29
> >
> > Thanks for the thorough rebuttal. However, I still have my original concerns about novelty, problem definition, performance, setting, and costs, i.e., the answers do not clearly resolve my questions. Therefore, I remain my original score.

---

> > > ### Author Response · Authors · 2024-11-30
> > >
> > > We sincerely appreciate your thorough review and continued engagement with our manuscript.
> > >
> > > In your recent feedback, you indicated that our responses did not clearly resolve your original concerns. To better serve the review process, we would greatly appreciate if you could elaborate on specific parts of our responses which require additional clarification or correction. We believe that this detailed discussion will substantially enhance the quality of our work.
> > >
> > > Thank you for your continued support and dedication to improving our research.

---

### Author Response · Authors · 2024-11-23
**[Common Response]**

We sincerely thank all reviewers for their thorough examination of our manuscript and their constructive feedback. We particularly appreciate the shared suggestion to strengthen the presentation of our contributions and better articulate the theoretical foundations underlying our algorithm's strong performance. To address this valuable feedback, we have prepared a comprehensive overview of our paper's contributions, which has been incorporated into the revised manuscript. We have attached a detailed list of our revisions below, with changes highlighted in the revised manuscript.

Our paper makes the following key contributions:

**Novel Framework Addressing Limitations in SL-based Heatmap Solvers.**

Our analysis reveals three fundamental issues in existing supervised learning (SL) based heatmap solvers that have been insufficiently addressed in prior works, where the primary focus has been on solution imitation:

- Ignorance of Cost in the Training Process
- Ignorance of Decoders in the Training Process
- Dependency on High-Quality Training Datasets

To address these issues, we propose what is, to our knowledge, the first reinforcement learning (RL) fine-tuning framework for CO that enables SL-based heatmap solvers to consider both decoder behavior and solution costs during training.

**Practical Algorithm Design for Large-scale Heatmap Solvers**.

Despite RL's natural fit for combinatorial optimization through direct cost minimization, scaling it to complex CO problems remains challenging. While RL has succeeded in autoregressive settings with well-defined MDPs, RL-based heatmap solvers like DIMES [1] underperform compared to SL-based methods. Alternative methods incorporating cost information during inference (e.g., MCTS, cost-guided search in T2T [2]) heavily depend on base model quality.

To bridge this performance gap and enable effective direct RL training of heatmap solvers, we carefully design our algorithm to overcome issues of previous works:

- A two-stage learning approach that combines SL pretraining and RL fine-tuning, achieving both training stability and performance gains
- A diffusion-based framework that excels as an SL-based heatmap solver while naturally providing a well-defined multi-step MDP through its denoising process, enabling effective RL application
- Novel training stability techniques, including Low Rank Adaptation and selective parameter freezing, to enhance RL fine-tuning reliability

**Comprehensive Empirical Validation**.

Through rigorous experimental analysis newly added in this revision, we systematically demonstrate our motivation, the three limitations of existing SL-based solvers, with quantitative evidence. Our experimental results validate that our proposed algorithms effectively address these limitations.

Our method consistently achieves superior performance compared to state-of-the-art Neural CO solvers across diverse CO benchmarks, including recent cost-aware approaches such as T2T and DIMES. This comprehensive empirical evaluation substantiates the effectiveness of our technical contributions.

**Conclusion**.

While our framework is built upon simple components, each element is meticulously designed to address critical limitations unique to combinatorial optimization. Our method demonstrates robust state-of-the-art performance across diverse CO benchmarks while maintaining consistent hyperparameters, highlighting its strong generalizability. This combination of architectural simplicity and superior empirical results not only validates our design choices but also suggests promising directions for adaptation to broader CO domains. We believe our contributions provide valuable insights for advancing research in heatmap-based CO solvers.

[1] Qiu, Ruizhong, Zhiqing Sun, and Yiming Yang. "Dimes: A differentiable meta solver for combinatorial optimization problems." *Advances in Neural Information Processing Systems* 35 (2022): 25531-25546.

[2] Li, Yang, et al. "From distribution learning in training to gradient search in testing for combinatorial optimization." *Advances in Neural Information Processing Systems* 36 (2024).

**Paper modification.**

The changes made to the paper are highlighted in blue text.

[modified] Clarified the contribution points at the end of the Introduction.

[modified] Moved the experiment in the Introduction (Figure 1) to the experiments section (Figure 3) for better alignment with the paper's message.

[Added] Briefly explained how the issues raised in Section 4.1 can be addressed using the proposed method.

[modified] Revised the explanation at the end of Section 4.2 to avoid misunderstanding about not using gradient-based guidance.

[Added] Provided a detailed explanation of the differences between T2T and the proposed work in Section 4.2 and Appendix F.

[modified] Moved transfer learning and TSPLIB experiments to the appendix due to space limitations.

[Added] Appendix C.2: Experiments with various fine-tuning methods.

---

### Meta-Review · Area_Chair_cGh1 · 2024-12-19

**Metareview:**

The paper titled "CADO: Cost-Aware Diffusion Models for Combinatorial Optimization via RL Fine-tuning" introduces CADO, a novel framework that integrates Reinforcement Learning (RL) fine-tuning with diffusion models to address Combinatorial Optimization (CO) problems such as the Traveling Salesman Problem (TSP) and Maximal Independent Set (MIS). The authors propose a two-phase training approach where a diffusion model is first trained using supervised learning (SL) and subsequently fine-tuned with RL to incorporate cost-awareness and improve solution quality. Empirical evaluations demonstrate that CADO achieves superior performance compared to state-of-the-art methods like DIFUSCO and T2T, showcasing reductions in computational costs and enhancements in solution accuracy across various benchmarks.

Despite its promising results, the paper exhibits several critical weaknesses that undermine its suitability for acceptance. The methodological innovations are perceived as incremental, primarily extending existing RL and diffusion model techniques without introducing substantial novel contributions. Comparisons with closely related methods, particularly T2T, are insufficiently addressed, raising concerns about the relative significance of CADO’s improvements. Additionally, the experimental evaluations, while comprehensive, reveal minimal performance gaps and inconsistent results across different datasets, which question the robustness and generalizability of the proposed approach. The reliance on multiple hyperparameters and the lack of clarity in explaining key design choices further obscure the paper’s contributions, making it difficult to assess the true impact and scalability of CADO in diverse CO scenarios.

**Additional Comments On Reviewer Discussion:**

During the rebuttal period, the authors made concerted efforts to address the reviewers’ concerns. Reviewer k7qD highlighted issues related to the novelty of the method, problem definition, and the minimal performance improvements, questioning whether the cost-awareness introduced by RL fine-tuning offered significant advantages over existing approaches like T2T and DIFUSCO. The authors responded by clarifying their methodological contributions, emphasizing the integration of RL during training rather than inference, and provided additional experimental results to demonstrate the effectiveness of CADO under various settings. However, Reviewer k7qD remained unconvinced, maintaining concerns about the method’s incremental nature and insufficient novelty.

Reviewer waxg pointed out the lack of detailed comparisons with peer methods, particularly T2T, and questioned the technical challenges addressed by combining RL with diffusion models. The authors responded by elaborating on the distinctions between CADO and T2T, providing additional comparative analyses and clarifying the technical contributions. Despite these clarifications, Reviewer waxg maintained a marginally above-threshold score, indicating lingering doubts about the significance of the contributions and the robustness of the experimental validations.

---

### Decision · Program_Chairs · 2025-01-22

Reject